# CLImage: Human-Annotated Datasets for Complementary-Label Learning

**Hsiu-Hsuan Wang, Tan-Ha Mai, Nai-Xuan Ye, Wei-I Lin, Hsuan-Tien Lin**
National Taiwan University
{b09902033, d10922024, b09902008, r10922076, htlin}@csie.ntu.edu.tw

## Abstract

Complementary-label learning (CLL) is a weakly-supervised learning paradigm that aims to train a multi-class classifier using only complementary labels, which indicate classes to which an instance does not belong. Despite numerous algorithmic proposals for CLL, their practical applicability remains unverified for two reasons. Firstly, these algorithms often rely on assumptions about the generation of complementary labels, and it is not clear how far the assumptions are from reality. Secondly, their evaluation has been limited to synthetic datasets. To gain insights into the real-world performance of CLL algorithms, we developed a protocol to collect complementary labels from human annotators. Our efforts resulted in the creation of four datasets: CLCIFAR10, CLCIFAR20, CLMicroImageNet10, and CLMicroImageNet20, derived from well-known classification datasets CIFAR10, CIFAR100, and TinyImageNet200. These datasets represent the very first real-world CLL datasets. Through extensive benchmark experiments, we discovered a notable decrease in performance when transitioning from synthetic datasets to real-world datasets. We investigated the key factors contributing to the decrease with a thorough dataset-level ablation study. Our analyses highlight annotation noise as the most influential factor in the real-world datasets. In addition, we discover that the biased-nature of human-annotated complementary labels and the difficulty to validate with only complementary labels are two outstanding barriers to practical CLL. These findings suggest that the community focus more research efforts on developing CLL algorithms and validation schemes that are robust to noisy and biased complementary-label distributions.

## 1 Introduction

Ordinary multi-class classification methods rely heavily on high-quality labels to train effective classifiers. However, such labels can be expensive and time-consuming to collect in many real-world applications. To address this challenge, researchers have turned their attention towards weakly-supervised learning, which aims to learn from incomplete, inexact, or inaccurate data sources [20, 28]. This learning paradigm includes but is not limited to noisy-label learning [5], partial-label learning [2], positive-unlabeled learning [3], and complementary-label learning [8].

In this work, we focus on complementary-label learning (CLL). This learning problem involves training a multi-class classifier using only complementary labels, which indicate the classes that a data instance does not belong to. Although several algorithms have been proposed to learn from complementary labels, they were only benchmarked on synthetic datasets with some idealistic

Submitted to the 38th Conference on Neural Information Processing Systems (NeurIPS 2024) Track on Datasets and Benchmarks. Do not distribute.

assumptions on complementary-label generation [1, 8, 9, 16, 21]. Thus, it remains unclear how well these algorithms perform in practical scenarios.

In particular, current CLL algorithms heavily rely on the *uniform assumption* for generating complementary labels [8], which specifies that complementary labels are generated by uniformly sampling from the set of all possible complementary labels. To alleviate the restrictiveness of the uniform assumption, Yu et al. [27] considered a more general *class-conditional assumption*, where the distribution of the complementary labels only depends on its ordinary labels. These assumptions have been used in many subsequent works to generate the *synthetic complementary datasets* for examining CLL algorithms [1, 9, 16, 21, 25]. Although these assumptions simplify the design and analysis of CLL algorithms, it remains unknown whether these assumptions hold true in practice and whether violation of these assumptions will significantly affect the performance of CLL algorithms. In addition to the uniform or class-conditional assumptions, most existing studies implicitly assumes that the complementary labels are noise-free. That is, they do not mistakenly represent the ordinary labels. While some studies claim to be more robust to noisy complementary labels [14], they were only tested on synthetic scenarios. It remains unclear how noisy the real-world datasets are, and how such noise affects the performance of current CLL algorithms.

To understand how much the real-world scenario differs from the assumptions, we started by collecting the datasets CLCIFAR10 and CLCIFAR20, which are derived from the famous CIFAR datasets for ordinary multi-class classification [12]. Since their release in 2023, the datasets [22] have been utilized by several emerging CLL studies [15, 23, 24, 26], demonstrating their instantaneous impact. We continue to extend the collection and form two additional human-annotated datasets, CLMicroImageNet10 and CLMicroImageNet20, which are derived from TinyImageNet200 [13, 19]. The extension verifies that our observations on CIFAR-derived datasets hold true for other image datasets. For all four datasets, we analyze the collected complementary labels, including their noise rates and non-uniform nature. Then, we perform benchmark experiments with diverse state-of-the-art CLL algorithms and conduct dataset-level ablation study on the assumptions of complementary-label generation using the collected datasets. Our studies reveal annotation noise as the most influential factor in the real-world datasets, and confirm that the non-uniform nature of human-annotated complementary labels cause certain CLL algorithms more susceptible to overfitting. These findings immediately suggest that the community focus more research efforts on developing CLL algorithms that are robust to noisy and non-uniform complementary-label distributions. In addition, we used the collected datasets to demonstrate that existing complementary-label-only validation schemes are not mature yet, suggesting the community a novel research direction for making CLL practical. Our contributions are summarized as follows:

- We designed a collection protocol of complementary labels (CLs) for images, and verified that the protocol collects reasonable human-annotated CLs across different datasets.

- We released **CLImage**, the collected set of four real-world CL datasets to support the continuous research of the community, publicly released at `https://github.com/ntucllab/CLImage_Dataset`.

- We analyzed the collected datasets with extensive benchmarking experiments, which provides novel and valuable insights for the community.

## 2 Preliminaries on CLL

### 2.1 Complementary-label learning

In ordinary multi-class classification, a dataset $D = \{(\mathbf{x}_i, y_i)\}_{i=1}^n$ that is *i.i.d.* sampled from an unknown distribution is given to the learning algorithm. For each $i$, $\mathbf{x}_i \in \mathbb{R}^M$ represents the $M$-dimension feature of the $i$-th instance and $y_i \in [K] = \{1, 2, \ldots, K\}$ represents the class $\mathbf{x}_i$ belongs to. The goal of the learning algorithm is to learn a classifier from $D$ that can predict the labels of unseen instances correctly. The classifier is typically parameterized by a scoring function $\mathbf{g} \colon \mathbb{R}^M \to \mathbb{R}^K$, and the prediction is made by $\arg\max_{k \in [K]} \mathbf{g}(\mathbf{x})_k$ given an instance $\mathbf{x}$, where $\mathbf{g}(\mathbf{x})_k$

denotes the $k$-th output of $\mathbf{g}(\mathbf{x})$. In contrast to ordinary multi-class classification, CLL shares the same goal of learning a classifier but trains with different labels. In CLL, the ordinary label $y_i$ is not accessible to the learning algorithm. Instead, a complementary label $\bar{y}_i$ is provided, which is a class that the instance $\mathbf{x}_i$ does *not* belong to. The goal of CLL is to learn a classifier that is able to predict the correct labels of unseen instances from a complementary-label dataset $\bar{D} = \{(\mathbf{x}_i, \bar{y}_i)\}_{i=1}^{n}$.

## 2.2 Common assumptions on CLL

Researchers have made some additional assumptions on the generation process of complementary labels to facilitate the analysis and design of CLL algorithms. One common assumption is the *class-conditional assumption* [27]. It assumes that the distribution of a complementary label only depends on its ordinary label and is independent of the underlying example's feature, i.e., $P(\bar{y}_i \mid \mathbf{x}_i, y_i) = P(\bar{y}_i \mid y_i)$ for each $i$. One special case of the class-conditional assumption is the *uniform assumption*, which further specifies that the complementary labels are generated uniformly. That is, $P(\bar{y}_i = k | y_i = j) = \frac{1}{K-1}$ for all $k \in [K] \backslash \{j\}$ [8, 9, 14].

For convenience, a $K \times K$ matrix $T$, called *transition matrix*, is often used to represent how the complementary labels are generated under the class-conditional assumption. $T_{j,k}$ is defined to be the probability of obtaining a complementary label $k$ if the underlying ordinary label is $j$, i.e., $T_{j,k} = P(\bar{y} = k \mid y = j)$ for each $j, k \in [K]$. The diagonals of $T$ hold the conditional probabilities that a complementary label mistakenly represents the ordinary label. That is, they indicate the noise level of the complementary labels. When $T$ contains all zeros on its diagonals, the CLL scenario is called *noiseless*. For instance, the uniform and noiseless assumption can be represented by $T_{j,j} = 0$ for each $j \in [K]$ and $T_{j,k} = \frac{1}{K-1}$ for each $k \neq j$. Class-conditional CLL scenarios based on any other non-uniform $T$ are often called *biased*.

## 2.3 A brief overview of CLL algorithms

The pioneering work by Ishida et al. [8] studied how to learn from complementary labels under the *uniform assumption* by converting the risk estimator in ordinary multi-class classification to an unbiased risk estimator (**URE**) in CLL [8]. **URE** is then found to be prone to overfitting because of negative empirical risks, and is upgraded with two tricks, non-negative risk estimator (**URE-NN**) and gradient accent (**URE-GA**) [9]. The *surrogate complementary loss* (**SCL**) algorithm mitigates the overfitting issue of **URE** by a different loss design that decreases the variance of the empirical estimation. However, these algorithms either rely on the uniform assumption in design or are only tested on the synthetic datasets that obeys the uniform assumption.

To make CLL one step closer to practice, researchers have explored algorithms to go beyond the uniform (and thus noiseless) assumption. Yu et al. [27] utilized the forward-correction loss (**FWD**) to accommodate biased complementary label generation by adapting techniques from noisy label learning [18] to change the loss. Additionally, Gao and Zhang [6] proposed the **L-W** algorithm based on discriminatively modeling the distribution of complementary labels through a weighting function, further improving the performance in bias scenario. Furthermore, Ishiguro et al. [10] designed robust loss functions for learning from noisy CLs, including **MAE** and **WMAE**, by applying the gradient ascent technique [9] to handle noisy scenarios.

Besides CLL algorithms, a crucial component for making CLL practical is model validation. In ordinary-label learning, this can be done by naively calculating the classification accuracy on a validation dataset. In CLL, this scheme can be intractable if there are not enough ordinary labels. One generic way of model validation is based on the result of Ishida et al. [9] by calculating the unbiased risk estimator of the zero-one loss, i.e.,

$$\hat{R}_{01}(\mathbf{g}) = \frac{1}{N} \sum_{i=1}^{N} e_{\bar{y}_i}^{\top}(T^{-1}) \ell_{01}(\mathbf{g}(x_i)) \tag{1}$$

where $e_{\bar{y}_i}$ denotes the one-hot vector of $\bar{y}_i$, $\ell_{01}(\mathbf{g}(x_i))$ denotes the $K$-dimensional vector $(\ell_{01}(\mathbf{g}(x_i), 1), \ldots, \ell_{01}(\mathbf{g}(x_i)), K))^T$, and $\ell_{01}(\mathbf{g}(x_i), k) = 0$ if $\arg\max_{k \in [K]} \mathbf{g}(x_i) = k$ and 1 oth-

erwise, representing the zero-one loss of $\mathbf{g}(x_i)$ if the ordinary label is $k$. This estimator will be used in the experiments in Section 6. Another validation objective, surrogate complementary esimation loss (SCEL), was proposed by Lin and Lin [14]. SCEL measures the log loss of the complementary probability estimates induced by the probability estimates on the ordinary label space. The formula to calculate SCEL is as follows,

$$\hat{R}_{\text{SCEL}}(\mathbf{g}) = \frac{1}{N} \sum_{i=1}^{N} -\log\left(e_{\bar{y}_i}^{\top} T^{\top} \text{softmax}(\mathbf{g}(x_i))\right). \tag{2}$$

## 3  Construction of the CLImage collection

In this section, we introduce the four complementary-labeled datasets that we collected, CLCIFAR10, CLCIFAR20, CLMicroImageNet10 and CLMicroImageNet20. All datasets are labeled by human annotators on Amazon Mechanical Turk (MTurk)[1].

### 3.1  Datasets and goals

The complementary-labeled datasets are derived from ordinary multi-class classification datasets. CIFAR10, CIFAR100 and TinyImageNet200 [12, 13, 19]. This selection is motivated by the real-world noisy label dataset by Wei et al. [25]. Building upon the CIFAR and TinyImageNet200 datasets allow us to estimate the noise rate and the empirical transition matrix easily, as they already contain nearly noise-free ordinary labels. In addition, many of the state-of-the-art CLL algorithms have been benchmarked on synthetic complementary labels with the CIFAR datasets [4, 11, 17]. Our CLCIFAR counterparts immediately allow a fair comparison to those results with the same network architecture.

In addition to our CLCIFAR extensions, we are the first to introduce (Tiny)ImageNet-derived datasets to the CLL literature. Such datasets serve two purposes. First, it allows us to confirm the validity of our collection protocol and findings beyond CIFAR-derived datasets. Second, ImageNet knowingly contains images of higher complexity than CIFAR and can thus be used to challenge the ability of existing CLL algorithms more realistically.

There is a historical note that is worth sharing with the community: We initially attempted to collect complementary labels based on the 100 classes in CIFAR100. But some preliminary testing soon revealed that state-of-the-art CLL algorithms cannot produce meaningful classifiers for 100 classes even on synthetic complementary labels that are uniformly and noiselessly generated. We thus set our collection goals to be 10-class classification, which is the focus of most current CLL studies, and 20-class classification, which extends the horizon of CLL and matches the 20 super-class structure in CIFAR.

### 3.2  Complementary label collection protocol

To collect only complementary labels from the CIFAR, TinyImageNet datasets, for each image in the training split, we first randomly sample four distinct labels and ask the human annotators to select any of the *incorrect* one from them. To leave room for analyzing the annotators' behavior, each image is labeled by three different annotators. The four labels are re-sampled for each annotator on each image. That is, each annotator possibly receives a different set of four labels to choose from. An algorithmic description of the protocol is as follows. For each image $\mathbf{x}$,

1. Uniformly sample four labels without replacement from the label set $[K]$.

2. Ask the annotator to select any one of the complementary label $\bar{y}$ from the four sampled labels.

3. Add the pair $(\mathbf{x}, \bar{y})$ to the complementary dataset.

---

[1]https://www.mturk.com/

169 Note that if the annotators always select one of the correct complementary labels uniformly, the
170 empirical transition matrix will also be uniform in expectation. We will inspect the empirical transition
171 matrix in Section 4. The labeling tasks are deployed on MTurk by dividing them into smaller we first
172 divide the total images into smaller human intelligence tasks (HITs). For instance, for constructing
173 the CLCIFAR datasets, we first divide the 50,000 images into five batches of 10,000 images. Then,
174 each batch is further divided into 1,000 HITs with each HIT containing 10 images. Each HIT is
175 deployed to three annotators, who receive 0.03 dollar as the reward by annotating 10 images. To
176 make the labeling task easier and increase clarity, the size of the images are enlarged to $200 \times 200$
177 pixels.

## 4 Result analysis

179 Next, we closely examine the collected complementary labels. We first analyze the error rates of the
180 collected labels, and then verify whether the transition matrix is uniform or not. Finally, we end with
181 an analysis on the behavior of the human annotators observed in the label collection protocol.

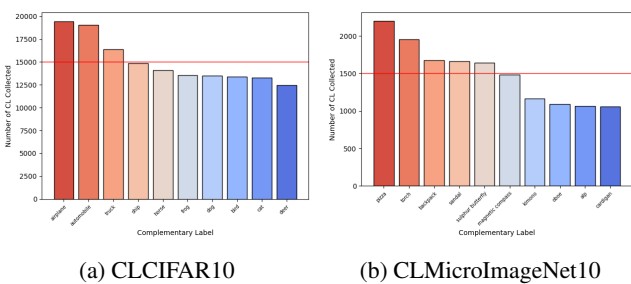

(a) CLCIFAR10      (b) CLMicroImageNet10

Figure 1: The label distribution of CLCIFAR10 and CLMicroImageNet10 datasets.

182 **Observation 1: noise rate compared to ordinary label collection** We first look at the noise rate of
183 the collected complementary labels. A complementary label is considered to be incorrect if it is actu-
184 ally the ordinary label. The mean error rate made by the human annotators is 3.93% for CLCIFAR10,
185 2.80% for CLCIFAR20, 5.19% for CLMicroImageNet10 and 3.21% for CLMicroImageNet20. In
186 theory, we can estimate a random annotator achieves a noise rate of $\frac{1}{K}$ for complementary label
187 annotation and a noise rate of $\frac{K-1}{K}$ for ordinary label annotation. If we compare the human annotators
188 to a random annotator, then for CLCIFAR10, human annotators have 60.7% less noisy labels than
189 the random annotator whereas for CIFAR10-N, human anotators have 80% less noisy labels. This
190 demonstrates that human annotators are more competent compared to a random annotator in the
191 ordinary-label annotation. Similarly, human annotators have 44% less noise than a random annotator
192 for CLCIFAR20 and 73.05% less noise for CIFAR100N-coarse. This observation reveals that while
193 the absolute noise rate is lower in annotating complementary labels, it may be more difficult to be
194 competent against random labels than the ordinary label annotation.

195 **Observation 2: imbalanced complementary label annotation** Next, we analyze the distribution of
196 the collected complementary labels. The frequency of the complementary labels for the CLCIFAR10
197 and CLMicroImageNet10 (CLMIN10) datasets are reported in Figure 1. As we can see in the
198 figure, the annotators exhibit specific biases towards certain labels. For instance, in CLCIFAR10,
199 annotators prefer "airplane" and "automobile," while in CLMIN10, they prefer "pizza" and "torch". In
200 CLCIFAR10, the bias is towards labels in different categories, as vehicles ("airplane," "automobile")
201 versus animals ("cat", "bird"). In contrast, in CLMIN10, the bias is towards items that are easily
202 recognizable ("pizza" and "torch") and against those that are less familiar ("cardigan" or "alp").

203 **Observation 3: biased transition matrix** Finally, we visualize the empirical transition matrix using
204 the collected CLs in Figure 2. Based on the first two observations, we could imagine that the transition
205 matrix is biased. By inspecting Figure 2, we further discover that the bias in the complementary
206 labels are dependent on the true labels. For instance, in CLCIFAR10, despite we see more annotations
207 on airplane and automobile in aggregate, conditioning on the transportation-related labels ("airplane",

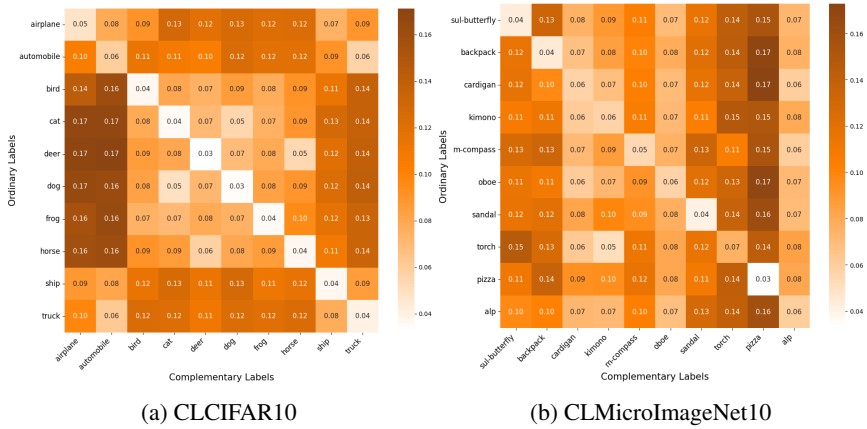

(a) CLCIFAR10             (b) CLMicroImageNet10

Figure 2: The empirical transition matrices of CLCIFAR10 and CLMicroImageNet10.

"automobile", etc), the distribution of the complementary labels becomes more biased towards other animal-related labels ("bird", "cat", etc.) Furthermore, this observation holds true on CLMIN10 as well. Next, we study the impact of the bias and noise on existing CLL algorithms.

We discovered similar patterns in all four human-annotated datasets, validating that our design methodology is practical for collecting real-world CLL image datasets. Due to space limitations, we have included the detailed analysis of CLCIFAR20 and CLMicroImageNet20 in Appendix B.4.

## 5 Experiments

In this section, we benchmarked several state-of-the-art CLL algorithms on CLImage. A significant performance gap between the models trained on the humanly annotated CLCIFAR, CLMicroImageNet dataset and those trained on the synthetically generated complementary labels (CL) was observed in Section 5.1, which motivates us to analyze the possible reasons for the gap with the following experiments. To do so, we discuss the effect of three factors in the label generating process, feature dependency, noise, and biasedness, in Section 5.2, Section 5.3, and Section 5.4, respectively. From our experiment results, we conclude that noise is the dominant factor affecting the performance of the CLL algorithms on CLCIFAR[2].

### 5.1 Standard benchmark on CLImage

**Baseline methods** Several state-of-the-art CLL algorithms were selected for this benchmark. Some of them take the transition matrix $T$ as inputs, which we call $T$-**informed** methods, including two version of forward correction [27]: **FWD-U** and **FWD-R**, two version of unbiased risk estimator with gradient ascent [9]: **URE-GA-U** and **URE-GA-R**, and robust loss [10] for learning from noisy CL: **CCE**, **MAE**, **WMAE**, **GCE**, and **SL**[3]. We also included some algorithms that assume the transition matrix $T$ to be uniform, called $T$-**agnostic** methods, including surrogate complementary loss **SCL-NL** and **SCL-EXP** [1], discriminative modeling **L-W** and its weighted variant (**L-UW**) [6], and pairwise-comparison (**PC**) with the sigmoid loss [8]. The details of the algorithms mentioned above are discussed in Appendix D.

**Implementation details** We collected and released three CLs per image to prepare for future studies. However, for this standard benchmark, we chose the first CL from the collected labels for each data instances to form a single CLL dataset, ensuring reproducibility. Then, we trained a ResNet18 [7] model using the baseline methods mentioned above on the single CLL dataset using

---

[2]Due to space and time constraints, we only provide the results and discussion on the CLCIFAR datasets.

[3]Due to space limitations, we only provided the results of MAE. The remaining results and discussions related to the robust loss methods can be found in Appendix B.3

Table 1: Standard benchmark results on CLCIFAR/ CLMicroImageNet(CLMIN) and uniform-CIFAR/ MicroImageNet(MIN) datasets. Mean accuracy ($\pm$ standard deviation) on the testing dataset from four trials with different random seeds. Highest accuracy in each column is highlighted in bold.

| | uniform-CIFAR10 | uniform-CIFAR20 | uniform-MIN10 | uniform-MIN20 | CLCIFAR10 | CLCIFAR20 | CLMIN10 | CLMIN20 |
|---|---|---|---|---|---|---|---|---|
| FWD-U | **64.19**±**0.57** | 21.54±0.37 | 36.30±1.12 | 12.57±2.94 | 34.83±0.50 | 8.03±0.74 | 23.85±2.76 | 6.33±1.04 |
| FWD-R | 61.32±0.90 | 21.50±0.38 | 35.70±1.19 | **14.85**±**1.75** | **38.13**±**0.88** | **20.27**±**0.53** | **30.15**±**1.83** | **10.60**±**0.82** |
| URE-GA-U | 50.24±1.11 | 16.67±1.35 | 35.70±1.97 | 11.65±1.90 | 34.72±0.40 | 10.49±0.52 | 22.90±2.97 | 5.75±0.43 |
| URE-GA-R | 50.73±1.83 | 17.57±0.61 | 33.65±1.40 | 9.78±3.88 | 30.23±0.70 | 6.17±0.82 | 13.25±5.11 | 6.50±0.35 |
| SCL-NL | 63.76±0.09 | 21.37±1.18 | **37.05**±**1.40** | 13.00±2.80 | 34.77±0.60 | 8.02±0.36 | 21.80±1.85 | 6.17±0.49 |
| SCL-EXP | 63.29±1.02 | **21.57**±**1.13** | 36.55±1.28 | 12.95±3.38 | 35.18±0.67 | 7.70±0.41 | 24.80±1.14 | 5.58±0.13 |
| L-W | 54.32±0.41 | 19.59±0.99 | 33.80±2.66 | 12.70±2.35 | 32.99±1.01 | 7.71±0.35 | 23.80±2.64 | 6.40±0.29 |
| L-UW | 57.52±0.59 | 20.71±0.92 | 35.10±2.74 | 12.12±3.13 | 34.69±0.32 | 8.15±0.30 | 22.40±1.67 | 6.35±0.86 |
| PC-sigmoid | 37.78±0.80 | 14.48±0.47 | 29.10±0.98 | 10.72±1.38 | 32.15±0.80 | 12.11±0.46 | 23.15±0.46 | 6.90±1.04 |
| ROB-MAE | 59.38±0.63 | 18.17±1.31 | 31.50±1.81 | 6.35±0.86 | 20.23±1.02 | 5.40±0.59 | 14.15±0.68 | 5.38±0.33 |

| | CIFAR10 | | CIFAR20 | | MIN10 | | MIN20 | |
|---|---|---|---|---|---|---|---|---|
| standard supervision | 82.80±0.28 | | 63.80±0.49 | | 68.70±1.53 | | 63.90±1.00 | |

the Adam optimizer for 300 epochs without learning rate scheduling. The weight decay was fixed at $10^{-4}$ and the batch size was set to 512. The experiments were run with Tesla V100-SXM2. For better generalization, we applied standard data augmentation technique, `RandomHorizontalFlip`, `RandomCrop`, and normalization to each image. The learning rate was selected from $\{10^{-3}, 5 \times 10^{-4}, 10^{-4}, 5 \times 10^{-5}, 10^{-5}\}$ using a 10% hold-out validation set. We selected the learning rate with the best classification accuracy on the validation dataset. Note that here we assumed the ordinary labels in the validation dataset are known. We will discuss other validation objectives that rely only on complementary labels in Section 6. As CLL algorithms are prone to overfitting [1, 9], some previous works did not use the model after training for evaluation. Instead, previous works were performed by evaluating the model on the validation dataset and selecting the epoch with the highest validation accuracy. In this work, we also follow the same aforementioned technique to validate testing set. For reference, we also performed the experiments on synthetically-generated CLL dataset, where the CLs were generated uniformly and noiselessly, denoted uniform-CIFAR.

**Results and discussion** As we can observe in Table 1, there is a significant performance gap between the humanly annotated dataset, CLCIFAR, and the synthetically generated dataset, uniform-CIFAR. The difference between the two datasets can be divided into three parts: (a) whether the generation of complementary labels depends on the feature, (b) whether there is noise, and (c) whether the complementary labels are generated with bias. A negative answer to those questions simplify the problem of CLL. We can gradually simplify CLCIFAR to uniform-CIFAR by chaining those assumptions as follows [4]:

$$\boxed{\text{CLCIFAR}} \xRightarrow[\text{Remove feature dependency}]{\text{Section 5.2}} \xRightarrow[\text{Remove noise}]{\text{Section 5.3}} \xRightarrow[\text{Remove biasedness}]{\text{Section 5.4}} \boxed{\text{uniform-CIFAR}}$$

In the following subsections, we will analyze how these three factors affect the performance of the CLL algorithms.

## 5.2 Feature dependency

In this experiment, we verified whether the performance gap resulted from the feature-dependent generation of practical CLs. Conceivably, even if two images belong to the same class, the distribution on the complementary labels could be different. On the other hand, the distributional difference could also be too small to affect model performance, e.g., if $P(\bar{y} \mid y, \mathbf{x}) \approx P(\bar{y} \mid y)$ for most $\mathbf{x}$. Consequently, we decided to further look into whether this assumption can explain the performance gap. To observe the effects of approximating $P(\bar{y} \mid y, \mathbf{x})$ with $P(\bar{y} \mid y)$, we generated two synthetic

---

[3]Note that FWD-R and URE-GA-R assume the empirical transition matrix $T_e$ to be provided. The empirical transition matrix is computed from the labels in the training set, so it is slightly different from a uniform transition matrix $T_u$ in the uniform-CIFAR datasets. As a result, the performances of FWD-R and URE-GA-R do not exactly match those of FWD-U and URE-GA-U, respectively, in the uniform-CIFAR datasets.

[4]The "interpolation" between CLCIFAR and uniform-CIFAR does not necessarily have to be this way. For instance, one can remove the biasedness before removing the noise. We chose this order to reflect the advance of CLL algorithms. First, researchers address the uniform case [8], then generalize to the biased case [27], then consider noisy labels [10]. There is no work considering feature-dependent complementary labels yet.

complementary datasets, CLCIFAR10-*iid* and CLCIFAR20-*iid* by i.i.d. sampling CLs from the empirical transition matrix in CLCIFAR10 and CLCIFAR20, respectively. We proceeded to benchmark the CLL algorithms on CLCIFAR-*iid* and presented the accuracy difference compared to CLCIFAR in Table 2.

**Results and discussion** From Table 2, we observed that the accuracy barely changes on the resampled CLCIFAR-*iid*, suggesting that even if the complementary labels in CLCIFAR could be feature-dependent, this dependency does not affect the model performance significantly. Hence, there might be other factors contributing to the performance gap.

Table 2: Mean accuracy difference ($\pm$ standard deviation) of different CLL algorithms. A plus indicates the performance on is calculated as CLCIFAR-*i.i.d.* accuracy minus CLCIFAR accuracy.

|  | FWD-U | FWD-R | URE-GA-U | URE-GA-R | SCL-NL | SCL-EXP | L-W | L-UW | PC-sigmoid |
|---|---|---|---|---|---|---|---|---|---|
| *CLCIFAR10-iid* | -1.1$\pm$2.17 | -0.36$\pm$1.15 | -3.03$\pm$1.25 | 0.74$\pm$0.35 | -0.67$\pm$1.81 | -1.97$\pm$1.16 | -2.5$\pm$0.56 | -3.53$\pm$1.36 | -2.03$\pm$2.05 |
| *CLCIFAR20-iid* | -0.64$\pm$0.39 | -3.53$\pm$1.13 | -0.37$\pm$0.51 | 1.79$\pm$2.34 | -0.28$\pm$0.61 | -0.39$\pm$0.69 | -0.5$\pm$1.37 | -0.82$\pm$0.04 | -2.24$\pm$0.52 |

## 5.3 Labeling noise

In this experiment, we further investigated the impact of the label noise on the performance gap. Specifically, we measured the accuracy on the noise-removed versions of CLCIFAR datasets, where varying percentages (0%, 25%, 50%, 75%, or 100%) of noisy labels are eliminated.

**Results and discussion** We present the performance of FWD trained on the noise-removed CLCIFAR10 dataset in the left figure in Figure 3. The results for other algorithms and the noise-removed CLCIFAR20 dataset can be found in Appendix E. From the figure, we observe a strong positive correlation between the performance and the proportion of removed noisy labels. When more noisy labels are removed, the performance gap diminishes and the accuracy approaches that of the ideal uniform-CLFAR dataset. Therefore, we conclude that the performance gap between the humanly annotated CLs and the synthetically generated CLs are primarily attributed to the label noise.

## 5.4 Biasedness of complementary labels

To further study the biasedness of CL as a potential factor contributing to the performance gap, we removed the biasedness from the noise-removed CLCIFAR dataset and examined the resulting accuracy. Specifically, we introduced the same level of uniform noise in uniform-CIFAR dataset and reevaluated the performance of FWD algorithms.

**Results and discussion** The striking similarity between the two curves in the right figure in Figure 3 shows that the accuracy is significantly influenced by label noise, while the biasedness of CL has a negligible impact on the results. Furthermore, we observe that the accuracy difference between the results of the last epoch and the best accuracy of validation set (or early-stopping: **ES**) results becomes smaller when the model is trained on the uniformly generated CLs. That is, the $T$-informed methods are more prone to overfitting when there is a bias in the CL generation.

With the experiment results in Section 5.2, 5.3, and 5.4, we can conclude that the performance gap between humanly annotated CL and synthetically generated CL is primarily attributed to label noise. Additionally, the biasedness of CLs may potentially contribute to overfitting, while the feature-dependent CLs do not detrimentally affect performance empirically. It is worth noting that in the last row of Table 1, the MAE methods that can learn from noisy CL fails to generalize well in the practical dataset. These results suggest that more research on learning with noisy complementary labels can potentially make CLL more realistic.

# 6 Validation Objectives

Validation is a crucial component in applying CLL algorithms in practice. With the collection of the real-world datasets, we are now able to estimate the difference between using ordinary labels for validation (the common practice in existing CLL studies, as what we do in Section 5) and using complementary labels for validation.

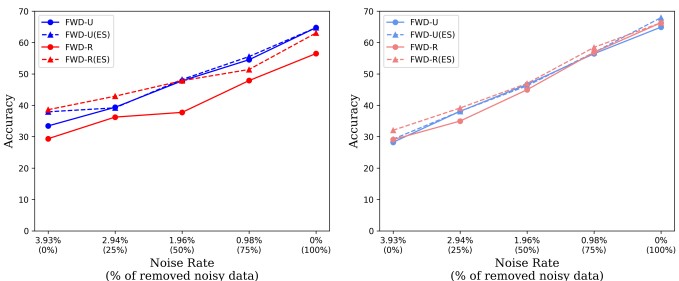

Figure 3: Accuracy of FWD-U and FWD-R on the noise-removed CLCIFAR10 dataset (**Left**) and the uniform-CIFAR10 dataset with uniform noise (**Right**) at varying noise rates.

Table 3: The testing accuracy of models evaluated with URE and SCEL.

| | CLCIFAR10 | | | | CLCIFAR20 | | | | CLMIN10 | | | | CLMIN20 | | | |
|---|---|---|---|---|---|---|---|---|---|---|---|---|---|---|---|---|
| | URE | SCEL | valid acc | gap (↓) | URE | SCEL | valid acc | gap (↓) | URE | SCEL | valid acc | gap (↓) | URE | SCEL | valid acc | gap (↓) |
| FWD-U | 33.13±1.30 | 31.86±1.52 | 34.83±0.50 | 1.70 | 6.70±0.46 | 7.10±0.48 | 8.03±0.74 | 0.93 | 20.75±2.12 | 20.20±0.72 | 23.85±2.76 | 3.10 | 4.97±0.72 | 4.55±0.81 | 6.33±1.04 | 1.35 |
| FWD-R | 33.70±3.38 | 35.64±1.37 | 38.13±0.88 | 2.49 | 17.35±2.32 | 18.40±1.56 | 20.27±0.53 | 1.86 | 22.15±4.15 | 29.15±1.93 | 30.15±1.83 | 1.00 | 8.60±1.32 | 9.90±1.19 | 10.60±0.82 | 0.70 |
| URE-GA-U | 30.45±3.58 | 33.21±1.12 | 34.72±0.40 | 1.51 | 7.03±0.61 | 8.71±0.74 | 10.49±0.52 | 1.79 | 17.05±3.35 | 21.30±3.01 | 22.90±2.97 | 1.60 | 4.27±0.80 | 5.03±0.48 | 5.75±0.43 | 0.72 |
| URE-GA-R | 27.39±1.89 | 28.32±1.38 | 30.23±0.70 | 1.91 | 3.58±0.47 | 5.42±0.96 | 6.17±0.82 | 0.75 | 8.90±1.03 | 10.30±1.53 | 13.25±5.11 | 2.95 | 5.15±0.62 | 5.57±1.54 | 6.50±0.35 | 0.93 |
| SCL-NL | 33.55±0.79 | 33.70±1.33 | 34.77±0.60 | 1.07 | 6.73±0.51 | 7.47±0.56 | 8.02±0.36 | 0.55 | 19.55±1.37 | 22.15±1.76 | 21.80±1.85 | -0.35 | 4.83±1.12 | 5.20±0.51 | 6.17±0.49 | 0.98 |
| SCL-EXP | 31.30±2.62 | 33.47±1.16 | 35.18±0.67 | 1.71 | 6.83±0.23 | 7.03±0.62 | 7.70±0.41 | 0.66 | 18.35±1.60 | 20.65±1.39 | 24.80±1.14 | 4.15 | 5.05±0.56 | 4.45±0.74 | 5.58±0.13 | 0.52 |
| L-W | 27.49±4.30 | 30.32±2.40 | 32.99±1.01 | 2.67 | 5.90±0.29 | 7.18±0.31 | 7.71±0.35 | 0.53 | 19.30±4.66 | 18.95±2.30 | 23.80±2.64 | 4.50 | 5.97±0.33 | 5.55±0.17 | 6.40±0.29 | 0.43 |
| L-UW | 28.90±2.01 | 29.78±2.69 | 34.69±0.32 | 4.91 | 6.40±0.42 | 8.16±0.30 | 8.15±0.30 | -0.01 | 18.25±4.31 | 19.80±1.61 | 22.40±1.67 | 2.60 | 5.82±0.77 | 6.48±1.03 | 6.35±0.86 | -0.13 |
| PC-sigmoid | 24.83±5.94 | 31.48±1.93 | 32.15±0.80 | 0.67 | 7.98±2.47 | 10.59±0.87 | 12.11±0.46 | 1.51 | 12.55±1.31 | 17.85±4.61 | 23.15±0.46 | 5.30 | 6.40±1.19 | 5.33±1.28 | 6.90±1.04 | 0.50 |
| ROB-MAE | 18.80±1.64 | 18.75±0.99 | 20.23±1.02 | 1.43 | 4.70±0.43 | 4.87±0.32 | 5.40±0.59 | 0.53 | 11.80±2.92 | 14.35±1.59 | 14.15±0.68 | -0.20 | 5.08±0.44 | 4.62±0.66 | 5.38±0.33 | 0.30 |

**Validation objectives** As discussed in Section 2, validating the model performance solely with complementary labels poses a non-trivial challenge. To the best of our knowledge, only two existing CLL studies offer some possibility to evaluate a classifier *with only complementary labels*. They are URE [9] and SCEL [14]. We take these two validation objectives to select the optimal learning rate from $\{10^{-3}, 5 \times 10^{-4}, 10^{-4}, 5 \times 10^{-5}, 10^{-5}\}$ and provides the accuracy on testing set in Table 3. We compare the result to another validation objective that computes the accuracy on an equal number of *ordinary labels*. Our goal was to determine the gap between using complementary labels and ordinary labels for validation. We selected the best learning rate based on the validation objectives for URE, SCEL, and ordinary-label accuracy, and then report the test performance, as shown in Table 3 for real-world datasets and Table 4 in the Appendix for synthetic datasets.

**Results and discussion** Firstly, there appears no clear winner between URE and SCEL, both using only CLs for validation. Validating with the ordinary-label accuracy generally provides stronger performance than URE/SCEL, and the test performance gap between validating with ordinary labels and validating with complementary labels can be as big as nearly 5%. These findings suggest that using purely complementary labels for validation, whether through URE or SCEL, still suffers from a non-negligible performance drop compared to using ordinary validation. That is, the numbers reported in existing studies, which validates with ordinal labels, can be optimistic for practice. Whether this gap can be further reduced remains an open research problem and the community can pay more attention on that to make CLL more practical.

# 7 Conclusion

In this paper, we devised a protocol to collect complementary labels from human annotators. Utilizing this protocol, we curated four real-world datasets, CLCIFAR10, CLCIFAR20, CLMicroImageNet10, and CLMicroImageNet20 and made them publicly available to the research community. Through our meticulous analysis of these datasets, we confirmed the presence of noise and bias in the human-annotated complementary labels, challenging some of the underlying assumptions of existing CLL algorithms. Extensive benchmarking experiments revealed that noise is a critical factor that undermines the effectiveness of most existing CLL algorithms. Furthermore, the biased complementary labels can trigger overfitting, even for algorithms explicitly designed to leverage this bias information. In addition, our study on the validation objective for CLL suggests that validating with only complementary labels causes significant performance degrading. These findings emphasize the need for the community to dedicate more effort on those issues. The curated datasets pave the way for the community to create more practical and applicable CLL solutions.

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
