# Appendix

## A    Limitations

To ensure the compatibility with previous CLL algorithms, our work focuses on image datasets based on CIFAR10/100, and TinyImageNet. It is worth investigating the real-world CLL datasets on larger datasets, such as ImageNet, and other domains. On the other hand, the proposed protocol focuses on collecting real-world complementary labels for analyzing the common assumptions on CLL. That said, it is also crucial to understand efficient ways to collect complementary labels in practice, e.g., by asking annotators binary questions to collect ordinary and complementary labels simultaneously. We leave these directions as future works and hope that our work can open the way for the community to understand these questions.

## B    More discussion on practical noise and extended ablation study

Our work found out that the labeling noise is the main factor contributing to the performance gap between synthetic CL and practical CL. Hence, we conducted deeper investigation into some directions to handle the practical noise. In Section B.1, we discussed the performance improvement when more human-annotated complementary labels were available. In Section B.2, we designed the synthetic CLCIFAR-N dataset to study the difference between synthetic uniform noise and practical noise. In Section B.3, we provided the benchmark results of all robust loss methods to emphasize the essence of studying a practical complementary label dataset. In Section B.4, we discussed result analysis of CLCIFAR20 and CLMicroImageNet20 datasets and described the process how MicroImageNet10 and MicroImageNet20 datasets generation in Section B.5.

### B.1    Multiple complementary labels

In this experiment, we studied the case when there were multiple CLs for a data instance. We duplicated the data instance and assigned them with another practical label from the annotators. The results of this experiment were summarized in Table 5.

For CLCIFAR10, we observe that the model achieved better learning performance when trained on data instances with more CLs. However, the issue of overfitting persists even with the increased number of labels. In the case of CLCIFAR20, we found that without employing early stopping techniques, it is challenging to achieve improved results as the number of labels increased. Furthermore, the overfitting problem becomes more pronounced with the increased number of labels. Overall, these findings shed light on the challenges posed by multiple CLs and the persistence of overfitting.

### B.2    Benchmarks with synthetic noise

**Generation process of CLCIFAR-N**    Inspired by the conclusions drawn in Section 5.3, we investigated another avenue of research: the generalization capabilities of methods when transitioning from synthetic datasets with uniform noise to practical datasets. To obtain a general synthetic dataset with minimum assumption, we introduced CLCIFAR-N. This synthetic dataset contains unifrom CL and uniform real world noise from CLCIFAR dataset. The complementary labels of CLCIFAR-N are *i.i.d.* sampled from $T_{syn}$, where the diagonal entries are set to be $3.93\%/10$ (for generating CL for CIFAR10) or $2.8\%/20$ (for generating CL for CIFAR20). The non-diagonal entries are uniformly distributed. This construction allows us to generate a synthetic dataset that mimics real-world scenarios more closely with minimum knowledge.

**Benchmark results**    We ran the benchmark experiments with the identical settings as in Section 5.1 and present the results in Table 6. The performance difference between sythetic noise and practical noise are illustrated in the *diff* columns. A smaller difference indicates a better generalization capability of the models. Interestingly, the robust loss methods exhibit superiority on the synthetic CLCIFAR10-N dataset but struggle to generalize well on real-world datasets. This finding suggests the existence of fundamental differences between synthetic noise and practical noise. Further investigation into these differences is left as an avenue for future research.

Table 4: The testing accuracy of models evaluated with URE and SCEL.

| | uniform-CIFAR10 | | | | uniform-CIFAR20 | | | | uniform-MIN10 | | | | uniform-MIN20 | | | |
|---|---|---|---|---|---|---|---|---|---|---|---|---|---|---|---|---|
| | URE | SCEL | valid acc | gap (↓) | URE | SCEL | valid acc | gap (↓) | URE | SCEL | valid acc | gap (↓) | URE | SCEL | valid acc | gap (↓) |
| FWD-U | 53.41±5.51 | 50.36±3.25 | 64.19±0.57 | 10.78 | 16.73±2.29 | 16.52±2.61 | 21.54±0.37 | 4.81 | 33.65±2.84 | 33.20±3.16 | 36.30±1.12 | 2.65 | 10.10±2.66 | 9.15±1.68 | 12.57±2.94 | 2.47 |
| FWD-R | 52.55±4.06 | 49.17±3.11 | 61.32±0.90 | 8.77 | 18.29±0.39 | 16.61±2.65 | 21.50±0.38 | 3.21 | 32.15±3.40 | 33.10±2.03 | 35.70±1.19 | 2.60 | 12.72±3.28 | 11.57±2.91 | 14.85±1.75 | 2.12 |
| URE-GA-U | 48.68±1.11 | 49.29±1.67 | 50.24±1.11 | 0.95 | 15.23±2.35 | 16.09±1.23 | 16.67±1.35 | 0.58 | 28.10±5.24 | 34.35±2.39 | 35.70±1.97 | 1.35 | 8.53±1.55 | 8.52±1.38 | 11.65±1.90 | 3.12 |
| URE-GA-R | 50.49±1.21 | 50.25±1.57 | 50.73±1.83 | 0.25 | 15.68±1.35 | 16.12±0.95 | 17.57±0.61 | 1.45 | 29.85±4.73 | 34.10±1.90 | 33.65±1.40 | -0.45 | 7.15±2.13 | 7.12±2.42 | 9.78±3.88 | 2.63 |
| SCL-NL | 54.32±6.71 | 51.03±3.12 | 63.76±0.09 | 9.44 | 15.65±3.06 | 16.32±3.11 | 21.37±1.18 | 5.05 | 32.95±3.13 | 33.20±3.69 | 37.05±1.40 | 3.85 | 11.50±3.76 | 9.28±2.55 | 13.00±2.80 | 1.50 |
| SCL-EXP | 50.98±6.83 | 41.61±3.52 | 63.29±1.02 | 12.30 | 16.71±2.72 | 16.15±2.55 | 21.57±1.13 | 4.86 | 32.95±2.91 | 29.70±2.83 | 36.55±1.28 | 3.60 | 10.53±2.02 | 8.83±3.19 | 12.95±3.38 | 2.43 |
| L-W | 46.88±9.44 | 50.36±0.47 | 54.32±0.41 | 3.95 | 16.26±1.93 | 14.67±1.59 | 19.59±0.99 | 3.33 | 17.70±9.90 | 28.60±5.15 | 33.80±2.66 | 5.20 | 8.58±1.25 | 7.70±0.35 | 12.70±2.35 | 4.12 |
| L-UW | 52.47±3.63 | 51.15±1.61 | 57.52±0.59 | 5.05 | 16.10±1.51 | 15.58±1.97 | 20.71±0.92 | 4.62 | 22.10±7.68 | 25.60±7.14 | 35.10±2.74 | 9.50 | 10.60±2.36 | 8.28±2.02 | 12.12±3.13 | 1.52 |
| PC-sigmoid | 35.29±1.67 | 34.82±1.24 | 37.78±0.80 | 2.49 | 13.41±0.95 | 13.40±0.72 | 14.48±0.47 | 1.07 | 25.55±5.99 | 27.05±5.66 | 29.10±0.98 | 2.05 | 7.75±1.73 | 8.72±0.26 | 10.72±1.38 | 2.00 |
| ROB-MAE | 57.99±1.72 | 57.79±2.03 | 59.38±0.63 | 1.39 | 17.07±2.02 | 15.62±1.79 | 18.17±1.31 | 1.11 | 30.15±4.22 | 29.15±2.90 | 31.50±1.81 | 1.35 | 5.42±0.27 | 5.03±0.54 | 6.35±0.86 | 0.92 |

Table 5: Learning with Multiple CL: The figure shows the classification accuracy of each task with early stopping indicated in brackets. The highest accuracy in each column is bolded for ease of comparison.

| | CLCIFAR10 | | | CLCIFAR20 | | |
|---|---|---|---|---|---|---|
| num CL | 1 | 2 | 3 | 1 | 2 | 3 |
| FWD-U | 34.09(36.83) | 41.95(41.53) | 42.88(45.18) | 7.47(8.27) | 8.28(8.78) | 8.15(10.27) |
| FWD-R | 28.88(**38.9**) | 34.33(**47.07**) | 37.84(**49.76**) | **16.14**(20.31) | **16.99**(23.41) | 15.54(**24.19**) |
| URE-GA-U | **34.59**(36.39) | **45.71**(44.85) | **45.97**(47.97) | 7.59(10.06) | 8.42(11.52) | 8.53(12.75) |
| URE-GA-R | 28.7(30.94) | 42.73(43.34) | 44.73(47.36) | 5.24(5.46) | 6.77(6.92) | 5.0(5.55) |
| SCL-NL | 33.8(37.81) | 40.67(42.58) | 43.39(45.2) | 7.58(8.53) | 6.77(6.92) | 5.0(5.55) |
| SCL-EXP | **34.59**(36.96) | 40.89(42.99) | 44.4(47.9) | 7.55(8.11) | 7.42(8.39) | 8.0(9.31) |
| L-W | 28.04(34.55) | 34.96(41.83) | 39.05(47.46) | 7.08(8.74) | 8.06(8.76) | 8.03(10.18) |
| L-UW | 30.63(35.13) | 38.05(43.32) | 39.49(45.82) | 7.36(8.71) | 7.03(8.55) | 7.86(10.11) |
| PC-sigmoid | 24.38(35.88) | 25.63(39.82) | 33.89(43.75) | 9.27(14.26) | 11.91(16.07) | **17.68**(14.13) |

## B.3   Results of the robust loss methods

The original design of the robust loss aims to obtain the optimal risk minimizer even in the presence of corrupted labels. However, their methods do not generalized well on practical datasets. The results are provided in Table 7. In other words, solely considering synthetic noisy CLs does not guarantee performance on real-world datasets. These results once again underscore the importance of the CLCIFAR dataset.

## B.4   Result analysis of CLCIFAR20 and MicroImageNet20

In this section, we further investigate the complementary labels collected from the CLCIFAR20 and MicroImageNet20 datasets. We followed similar observation and analyzed in the Section 4. Our observation and analysis are described as below:

**Observation 1: noise rate compared to ordinary label collection** We observed that the noise rates for the complementary labels collected from the CLCIFAR20 and MicroImageNet20 datasets are 2.80% and 3.21%, respectively. This finding is consistent with the observations discussed in Section 4. The lower noise rate in the CLCIFAR20 dataset compared to MicroImageNet20 can be attributed to the greater difficulty in labeling the MicroImageNet20 dataset.

**Observation 2: imbalanced complementary label annotation** Next, we analyzed the distribution of the collected complementary labels. The frequencies of these labels for the CLCIFAR20 and CLMicroImageNet20 (CLMIN20) datasets are shown in Figure 4. The figure reveals that annotators exhibit specific biases towards certain labels. For example, in CLCIFAR20, annotators show a preference for labels such as "fish", "flowers", "people", "trees", "food container", and "transportation vehicles". In CLMIN20, they favor "iPod" and "tractor". In CLCIFAR20, the bias tends towards labels with shorter, more concrete, and understandable names. Conversely, in CLMIN20, the preference is for easily recognizable items as "iPod", and "tractor", while less familiar items such as "bannister", "american lobster", "snorkel", and "gazelle" are less favored.

**Observation 3: biased transition matrix** Finally, we visualized the empirical transition matrix using the collected complementary labels, as shown in Figure 5. Our observations indicate that the transition matrix is biased. Specifically, we discovered that the bias in the complementary labels is dependent on the true labels, as depicted in Figure 5. In CLCIFAR20, there are more annotations for labels with shorter, more concrete, and understandable names, such as "fish," "flowers," "people," and "transportation vehicles." This results in a distribution that is more biased towards these labels. A

Table 6: Benchmark results on CLCIFAR-N datasets. The classification accuracy difference is calculated by subtracting the practical CLCIFAR dataset from the performance on the synthetic CLCIFAR-N dataset.

| | CLCIFAR10-N | diff(↓) | CLCIFAR20-N | diff(↓) |
|---|---|---|---|---|
| FWD-U | 37.1 | 2.2 | **7.58** | 0.11 |
| FWD-R | - | - | - | - |
| URE-GA-U | 31.29 | -3.3 | 8.1 | 0.5 |
| URE-GA-R | - | - | - | - |
| SCL-NL | 37.79 | 2.06 | 7.75 | 0.16 |
| SCL-EXP | 35.86 | 3.19 | 6.95 | -0.59 |
| L-W | 30.1 | 2.06 | 6.16 | -0.91 |
| L-UW | 32.69 | 2.05 | 6.89 | -0.47 |
| PC-sigmoid | 19.64 | -4.73 | 6.54 | -2.72 |
| CCE | 32.34 | 13.45 | 5.71 | 0.71 |
| MAE | **41.34** | 23.09 | 6.83 | 1.83 |
| WMAE | 37.62 | 22.26 | 6.36 | 1.08 |
| GCE | 35.00 | 18.71 | 6.7 | 1.7 |
| SL | 29.98 | 12.29 | 6.08 | 1.05 |

Table 7: Standard benchmark results on CLCIFAR and uniform-CIFAR datasets for the robust loss method. Mean accuracy (± standard deviation) on the testing dataset from four trials with different random seeds. Highest accuracy in each column is highlighted in bold.

| | uniform-CIFAR10 | | CLCIFAR10 | | uniform-CIFAR20 | | CLCIFAR20 | |
|---|---|---|---|---|---|---|---|---|
| methods | valid_acc | valid_acc (ES) | valid_acc | valid_acc (ES) | valid_acc | valid_acc (ES) | valid_acc | valid_acc (ES) |
| CCE | $46.57\pm1.75$ | $49.51\pm0.73$ | $16.18\pm2.97$ | $20.18\pm3.39$ | $12.54\pm0.40$ | $14.62\pm1.29$ | $5.07\pm0.05$ | $5.41\pm0.30$ |
| MAE | $57.37\pm0.48$ | $58.50\pm0.97$ | $16.30\pm2.27$ | $19.44\pm4.41$ | $16.72\pm1.52$ | $17.63\pm1.63$ | $5.11\pm0.11$ | $5.87\pm0.26$ |
| WMAE | - | - | $13.01\pm1.89$ | $15.51\pm0.75$ | - | - | $5.31\pm0.27$ | $6.65\pm0.65$ |
| GCE | $58.10\pm1.54$ | $59.44\pm2.30$ | $14.31\pm1.44$ | $18.97\pm2.16$ | $15.86\pm1.93$ | $17.09\pm1.19$ | $5.21\pm0.29$ | $5.76\pm0.32$ |
| SL | $41.13\pm1.64$ | $42.64\pm0.11$ | $16.45\pm2.80$ | $19.28\pm3.16$ | $13.60\pm0.55$ | $15.70\pm1.23$ | $5.44\pm0.29$ | $6.59\pm0.43$ |

similar pattern of bias is observed in CLMIN20, where annotators favored easily recognizable items like "iPod" and "tractor", while less familiar items received fewer annotations.

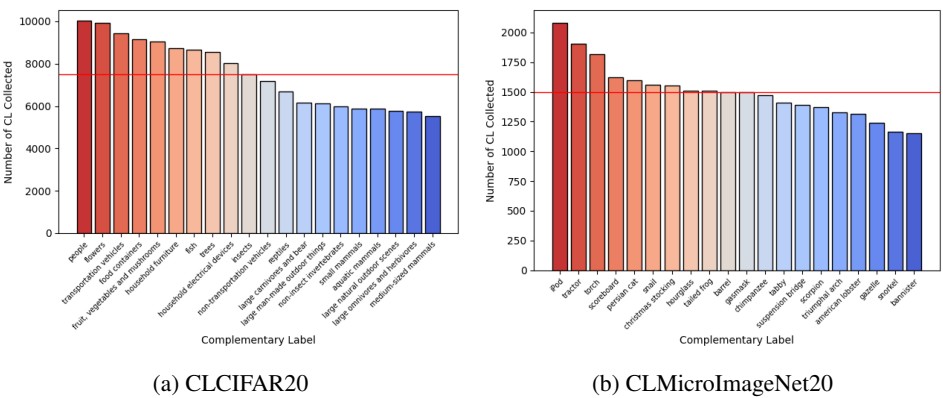

(a) CLCIFAR20      (b) CLMicroImageNet20

Figure 4: The label distribution of CLCIFAR20 and CLMicroImageNet20 datasets.

## B.5 MicroImageNet dataset generation

To generate the MicroImageNet10 and MicroImageNet20 datasets, we began by randomly selecting 10 classes from the 200 available in MicroImageNet to create MicroImageNet10. Similarly, we randomly selected 20 classes to form MicroImageNet20. The selected classes are listed in Table 10 of Appendix F. Each class in the TinyImageNet200 dataset contains multiple labels. To ensure reproducibility and facilitate human annotation, we chose the first label to represent the primary label of each class, as detailed in Appendix F. Each class in the MicroImageNet10/20 datasets comprises 500 images for the training set and 50 images for the validation set. To collect complementary labels for the MicroImageNet10/20 datasets, we followed a protocol similar to the one described in Section 3.2.

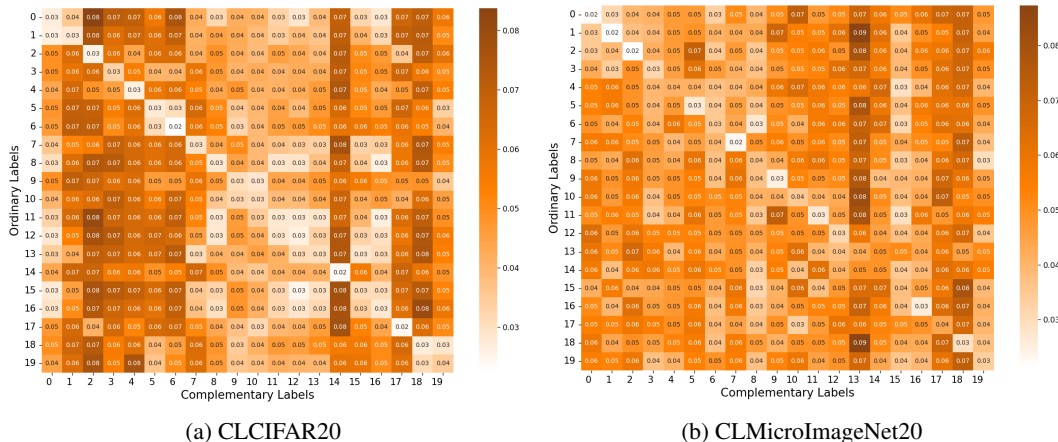

|         (a) CLCIFAR20         |    (b) CLMicroImageNet20    |

Figure 5: The empirical transition matrices of CLCIFAR20 and CLMicroImageNet20. The label names of CLCIFAR20 and CLMicroImageNet20 are abbreviated as indexes to save space. The full label names are provided in Appendix F.

Table 8: The overfitting results when there is no data augmentation.

|  | uniform-CIFAR10 | | CLCIFAR10 | | uniform-CIFAR20 | | CLCIFAR20 | |
|---|---|---|---|---|---|---|---|---|
| methods | valid_acc | valid_acc (ES) | valid_acc | valid_acc (ES) | valid_acc | valid_acc (ES) | valid_acc | valid_acc (ES) |
| FWD-U | **48.44** | **49.33** | 21.29 | 25.59 | **17.4** | **17.97** | 6.91 | 7.32 |
| FWD-R | - | - | 14.97 | **28.3** | - | - | 6.82 | **14.67** |
| URE-GA-U | 39.55 | 39.67 | 21.0 | 23.53 | 13.52 | 14.08 | 5.55 | 8.38 |
| URE-GA-R | - | - | 19.81 | 20.8 | - | - | 5.0 | 6.43 |
| SCL-NL | 48.2 | 48.27 | **21.96** | 26.51 | 16.55 | 17.54 | **7.1** | 7.92 |
| SCL-EXP | 46.79 | 47.52 | 21.89 | 27.66 | 16.18 | 17.89 | 6.9 | 7.3 |
| L-W | 27.02 | 44.78 | 20.06 | 27.6 | 10.39 | 16.3 | 5.64 | 8.02 |
| L-UW | 31.3 | 46.38 | 20.28 | 26.26 | 12.33 | 16.32 | 6.03 | 8.14 |
| PC-sigmoid | 18.97 | 33.26 | - | - | 7.67 | 10.41 | - | - |

## C  More discussion on biasedness

In addition to the label noise, the biasedness of CL in practical dataset would lead to overfitting, especially for those T-informed algorithms. We conducted deeper investigation into this phenomenon. In Section C.1, we demonstrated the necessity of employing data augmentation techniques to prevent overfitting. In Section C.2, we attempted to address the issue of overfitting by employing an interpolated transition matrix for regularization.

### C.1  Ablation on data augmentation

To further investigate the significance of data augmentation, we conducted identical experiments without employing data augmentation during the training phase. As we can observe in the training curves in Figure 6, data augmentation could improve the testing accuracy of all the algorithms we considered.

We also provide the results without the use of data augmentation techniques in Table 8, and we observed that almost all methods suffered from overfitting. It is worth noting that URE with gradient ascent suffers less compared to the other methods. The reason might be that reversing the gradient of the class with negative loss (the overfitting class) can be seen as a regularization technique. Therefore, URE with GA methods can be more resistant to overfitting in practical datasets.

### C.2  Ablation on interpolation between $T_u$ and $T_e$

In Table 1, we discovered that the $T$-informed methods did not always deliver better testing accuracy when $T_e$ is given. Looking at the difference between the accuracy of using early-stopping and not using early-stopping, we observe that when the $T_u$ is given to the $T$-informed methods, the difference becomes smaller. This suggests that $T$-informed methods using the empirical transition matrix has

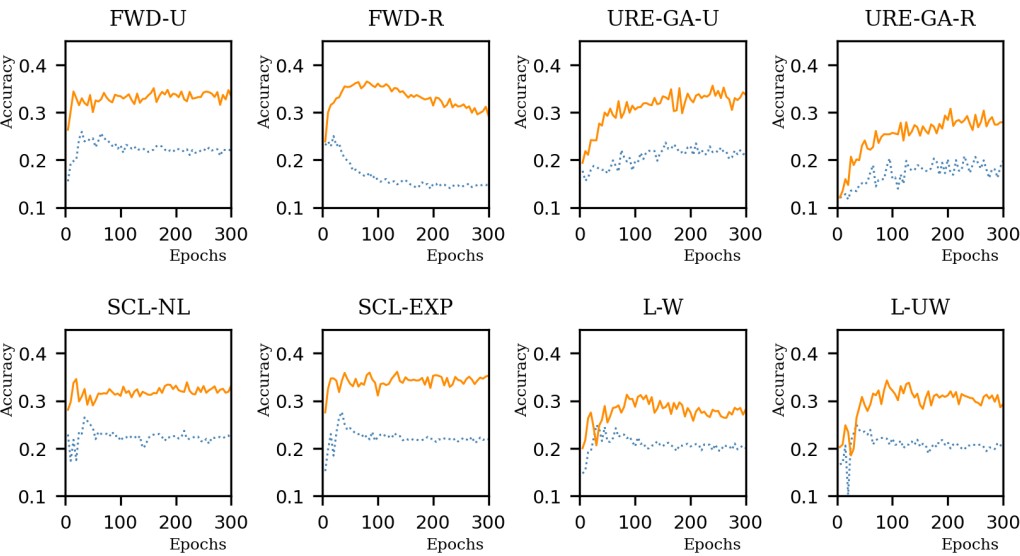

Figure 6: The Overfitting accuracy curve of FWD, URE, SCL-NL, L-W. The dotted line represents the accuracy obtained without data augmentation, while the solid line represents the accuracy with data augmentation included for reference. The accuracy of FWD, SCL-NL, SCL-EXP, L-W, L-UW methods reaches its highest at approximately the 50 epoches and converges to some lower point. The detail numbers are in Table 8

greater tendency to overfitting. On the other hand, $T$-informed methods using the uniform transition matrix could be a more robust choice.

We observe that the uniform transition matrix $T_u$ acts like a regularization choice when the algorithms overfit on CLCIFAR. This results motivate us to study whether we can interpolate between $T_u$ and $T_e$ to let the algorithms utilize the information of transition matrix while preventing overfitting. To do so, we provide an interpolated transition matrix $T_{\text{int}} = \alpha T_u + (1 - \alpha)T_e$ to the algorithm, where $\alpha$ controls the scale of the interpolation. As FWD is the $T$-informed method with the most sever overfitting when using $T_u$, we performed this experiment using FWD adn reported the results in Figure 7. As shown in Figure 7, FWD can learn better from an interpolated $T_{\text{int}}$, confirming the conjecture that $T_u$ can serve as a regularization role.

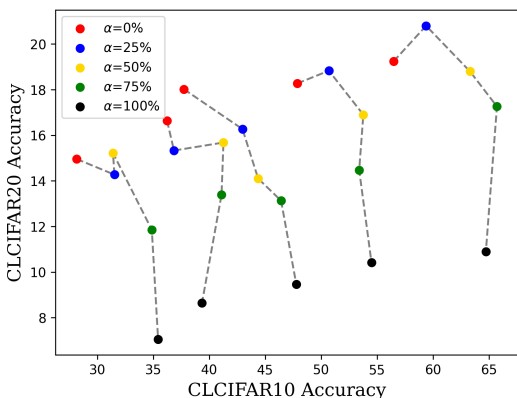

Figure 7: The last epoch accuracy of CLCIFAR10 and CLCIFAR20 for FWD algorithm with an $\alpha$-interpolated transition matrix $T_{\text{int}}$. The five solid points on each cruve represent different noise cleaning rate: 0%, 25%, 50%, 75%, 100% from left to right.

# D An overview of the complementary-label learning algorithms

In this section, we review the algorithms benchmarked in Section 5.

## D.1 T-informed CLL algorithms

Some of them take the transition matrix $T$ as inputs, which we call $T$-informed methods, including

- Two versions of forward correction method [28]: **FWD-U** and **FWD-R**. They utilize a uniform transition matrix $T_u$ and an empirical transition matrix $T_e$ as input, respectively.
- Two versions of unbiased risk estimator with gradient ascent [10]: **URE-GA-U** with a uniform transition matrix $T_u$ and **URE-GA-R** with an empirical transition matrix $T_e$.
- Robust loss methods [11] for learning from noisy CL, including **CCE**, **MAE**, **WMAE**, **GCE**, and **SL**[5]. We applied the gradient ascent technique [10] as recommended in the original paper.

In practice, the empirical transition matrix $T_e$ is not accessible to the learning algorithm, but we assume that the correct $T_e$ is given to **FWD-R**, **URE-GA-R** and the robust loss methods for simplicity.

T-informed CLL algorithms are those that has the transition matrix as inputs, includes but not limited to Forward loss correction (FWD) and Unbiased risk estimate (URE). They are expected to utilize the information of the transition matrix to provide better performance when the complementary labels are not generated uniformly. The transition matrix, however, may not be accessible in practice. In this case, a uniform transition matrix $T_u$ is typically provided to the algorithms as a default choice. In the benchmark in Section 5, we considered both scenarios in which the empirical transition matrix $T_e$ or the uniform transition matrix $T_u$ was provided.

**FWD** Forward loss correction utilizes the information of a transition matrix $T$ in its loss function as in Eq. 3 [28]. Essentially, this method trains model $f$ by minimizing the following loss function.

$$R(\mathbf{g}) = \frac{1}{N} \sum_{i=1}^{N} \ell(T^{\top} \operatorname{sm}(\mathbf{g}(x_i)), \bar{y}_i) \tag{3}$$

where $T$ is the transition matrix provided to the method and sm denotes the softmax function. We use **FWD-U** and **FWD-R** to indicate the cases that $T$ equals $T_u$ and $T_e$, respectively.

**URE-GA** Ishida et al. [9] proposed an unbiased risk estimator (URE) for learning from complementary label. The loss of the URE is defined as follows,

$$R(\mathbf{g}) = \frac{1}{N} \sum_{i=1}^{N} e_{\bar{y}_i}^{\top}(T^{-1}) \ell(\mathbf{g}(x_i)) \tag{4}$$

URE, however, can go below zero during the optimization procedure, leading to overfitting of the model. To address this issue, Ishida et al. [10] proposed two tricks, non-negative risk estimator (NN) and gradient accent(GA). The former zeros out the gradient when the mini-batch loss goes below zero while the latter reverse the mini-batch gradient when the loss from any of the complementary class goes below zero. We replace the transition matrix $T$ in the risk estimator 4 with $T_u$ and $T_e$ for **URE-GA-U** and **URE-GA-R**.

## D.2 T-agnostic CLL algorithms

T-agnostic CLL algorithms are those that do not take the information of the transition matrix, includes but not limited to Surrogate complementary loss (SCL) and Discriminative modeling (L-W/L-UW).

---

[5]Due to space limitations, we only provided the results of MAE. The remaining results and discussions related to the robust loss methods can be found in Appendix B.3.

**SCL**  Chou et al. [1] proposed to use the surrogate complementary loss (SCL) to address the overfitting tendency in URE. The loss function is defined as follows,

$$R(\mathbf{g}) = \frac{1}{N} \sum_{i=1}^{N} \phi(\bar{y}_i, \mathbf{g}(x_i)),\tag{5}$$

where $\phi(\cdot)$ is a surrogate loss for $0-1$ loss. For instance, SCL-NL uses the negative log loss $\phi(\bar{y}, \mathbf{g}(\mathbf{x})) = -\log(1 - p_{\bar{y}})$ and SCL-EXP uses the exponential loss $\phi(\bar{y}, \mathbf{g}(\mathbf{x})) = \exp(p_{\bar{y}})$.

**L-W/L-UW**  Gao and Zhang [7] proposed to use discriminative modeling to directly model the distribution of complementary labels. To do so, they proposed the following loss functions,

$$R(\mathbf{g}) = \frac{1}{N} \sum_{i=1}^{N} -\log(\mathrm{sm}(1 - \mathrm{sm}(\mathbf{g}(x))))_{\bar{y}_i},\tag{6}$$

where $\mathrm{sm}$ denotes the softmax function. They also proposed a weighting function to further improve the performance. The unweighted version is denoted as L-UW and the weighted version is denoted as L-W.

### D.3  Robust loss methods

Ishiguro et al. [11] studied two conditions on loss functions: weighted symmetric condition and relaxation of weighted symmetric condition. Five loss functions that can be robust against the estimation error of the transition matrix were proposed. Their results can be further generalized to noisy complementary label learning. More experiment details for reproduction can be found in their paper.

## E  Additional charts for CLCIFAR dataset with data cleaning

We remove 0%, 25%, 50%, 75%, 100% of the noisy data in CLCIFAR10 and CLCIFAR20 datasets. We discover that by removing the noisy data in the practical dataset, the practical performance gaps vanish for all the CLL algorithms. Therefore, we can conclude that the main obstacle to the practicality of CLL is label noise.

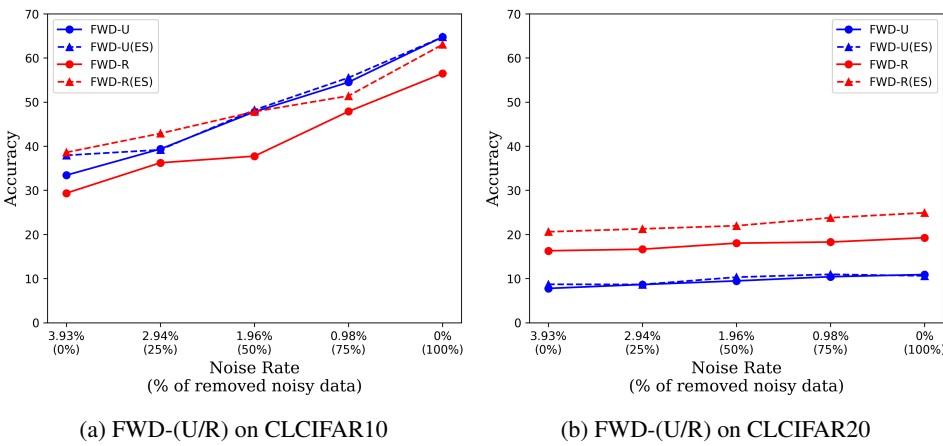

(a) FWD-(U/R) on CLCIFAR10          (b) FWD-(U/R) on CLCIFAR20

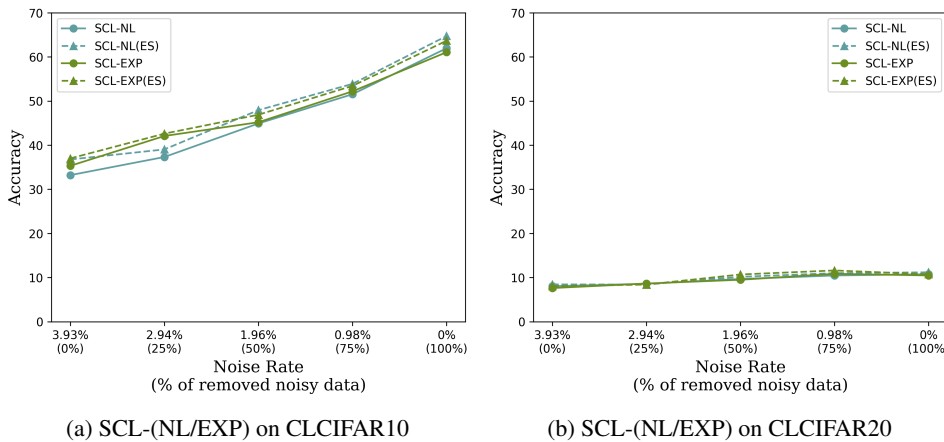

(a) SCL-(NL/EXP) on CLCIFAR10        (b) SCL-(NL/EXP) on CLCIFAR20

## F    Label names of CLCIFAR20 and CLMicroImageNet20

Table 9: The correspondence between index and label names of CLCIFAR20 and CLMicroImageNet20 datasets.

| Index | CLCIFAR20 Label Name | CLMicroImageNet20 Label Name |
|-------|----------------------|------------------------------|
| 0 | aquatic mammals | tailed frog |
| 1 | fish | scorpion |
| 2 | flowers | snail |
| 3 | food containers | american lobster |
| 4 | fruit, vegetables and mushrooms | tabby |
| 5 | household electrical devices | persian cat |
| 6 | household furniture | gazelle |
| 7 | insects | chimpanzee |
| 8 | large carnivores and bear | bannister |
| 9 | large man-made outdoor things | barrel |
| 10 | large natural outdoor scenes | christmas stocking |
| 11 | large omnivores and herbivores | gasmask |
| 12 | medium-sized mammals | hourglass |
| 13 | non-insect invertebrates | iPod |
| 14 | people | scoreboard |
| 15 | reptiles | snorkel |
| 16 | small mammals | suspension bridge |
| 17 | trees | torch |
| 18 | transportation vehicles | tractor |
| 19 | non-transportation vehicles | triumphal arch |

Table 10: The selected classes/folders for MicroImageNet10 (MIN10) and MicroImageNet20 (MIN20) are drawn from the TinyImageNet200 dataset. The labels provided in the table represent the **first** ordinary label for these classes.

| Index | MIN10 Folder | MIN10 Label Name | Index | MIN20 Folder | MIN20 Label Name |
|---|---|---|---|---|---|
| 0 | n02281406 | sulphur-butterfly | 0 | n01644900 | tailed frog |
| 1 | n02769748 | backpack | 1 | n01770393 | scorpion |
| 2 | n02963159 | cardigan | 2 | n01944390 | snail |
| 3 | n03617480 | kimono | 3 | n01983481 | american lobster |
| 4 | n03706229 | magnetic-compass | 4 | n02123045 | tabby |
| 5 | n03838899 | oboe | 5 | n02123394 | persian cat |
| 6 | n04133789 | scandal | 6 | n02423022 | gazelle |
| 7 | n04456115 | torch | 7 | n02481823 | chimpanzee |
| 8 | n07873807 | pizza | 8 | n02788148 | bannister |
| 9 | n09193705 | alp | 9 | n02795169 | barrel |
| | | | 10 | n03026506 | christmas stocking |
| | | | 11 | n03424325 | gasmask |
| | | | 12 | n03544143 | hourglass |
| | | | 13 | n03584254 | iPod |
| | | | 14 | n04149813 | scoreboard |
| | | | 15 | n04251144 | snorkel |
| | | | 16 | n04366367 | suspension bridge |
| | | | 17 | n04456115 | torch |
| | | | 18 | n04465501 | tractor |
| | | | 19 | n04486054 | triumphal arch |

## G  Analysis between multiple label collection trials

We carried out the same protocol for three independent trials to ensure the consistency of our results. The noise rates of CLCIFAR10 are 0.0398, 0.03882, and 0.03928 for three trials respectively. On the other hand, the noise rates of CLCIFAR20 are 0.02322, 0.02902, and 0.03196. These results show that the obtained noise rates are reliable and consistent. Besides, we also analyzed the distribution of complementary label within three trials as reported in Figure 10. The consistent distribution of complementary labels reveals the empirical human annotating biasedness within our protocol. Both analyses show that our protocol and discovery are solid and stable.

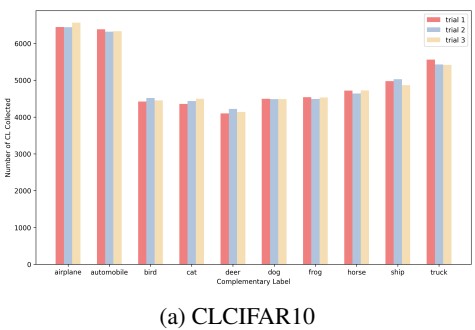

(a) CLCIFAR10

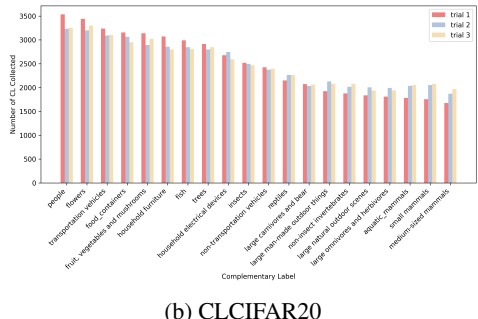

(b) CLCIFAR20

Figure 10: The complementary label distribution of three independent trials of CLCIFAR10 dataset (**Left**) and CLCIFAR20 dataset (**Right**).

## H  AutoAugment

In addition to the standard data augmentation, RandomCrop and RandomHorizontalFlip, we also considered a more advanced one, AutoAugment [3]. The benchmark results using AutoAugment are provided in Table 11. We observe that AutoAugment can improve the performance in almost all of the secenarios with a cost of around double running time compared to standard data augmentation. Also, the overfitting tendency of the previous algorithms remains unsolved although we observe that early-stopping can still deliver better performance when using AutoAugment.

Table 11: Comparison of performance using **AutoAugment** on CLCIFAR and uniform-CIFAR datasets in relation to tab:exp-1. The accuracy changes are shown in subscript, with enhanced accuracy values being highlighted in bold.

| | uniform-CIFAR10 | | CLCIFAR10 | | uniform-CIFAR20 | | CLCIFAR20 | |
| --- | --- | --- | --- | --- | --- | --- | --- | --- |
| methods | valid_acc | valid_acc (ES) | valid_acc | valid_acc (ES) | valid_acc | valid_acc (ES) | valid_acc | valid_acc (ES) |
| FWD-U | $\mathbf{75.72}_{+6.55}$ | $\mathbf{77.02}_{+7.23}$ | $\mathbf{42.46}_{+8.37}$ | $\mathbf{43.52}_{+6.69}$ | $\mathbf{26.8}_{+6.56}$ | $\mathbf{26.93}_{+6.31}$ | $7.38_{-0.09}$ | $\mathbf{8.76}_{+0.49}$ |
| FWD-R | $\mathbf{75.53}_{+5.79}$ | $\mathbf{76.06}_{+6.47}$ | $\mathbf{40.37}_{+11.49}$ | $\mathbf{42.13}_{+3.23}$ | $\mathbf{26.74}_{+6.74}$ | $\mathbf{26.98}_{+6.27}$ | $\mathbf{20.71}_{+4.57}$ | $\mathbf{24.77}_{+4.46}$ |
| URE-GA-U | $\mathbf{60.24}_{+5.62}$ | $\mathbf{59.9}_{+4.96}$ | $\mathbf{37.78}_{+3.19}$ | $\mathbf{38.48}_{+2.09}$ | $\mathbf{17.08}_{+1.67}$ | $\mathbf{18.59}_{+2.0}$ | $\mathbf{8.88}_{+1.29}$ | $9.7_{-0.36}$ |
| URE-GA-R | $\mathbf{58.36}_{+5.06}$ | $\mathbf{59.42}_{+2.40}$ | $\mathbf{31.98}_{+3.28}$ | $\mathbf{33.08}_{+2.14}$ | $\mathbf{18.2}_{+3.34}$ | $\mathbf{19.72}_{+2.39}$ | $\mathbf{10.85}_{+5.61}$ | $\mathbf{9.89}_{+4.43}$ |
| SCL-NL | $\mathbf{76.6}_{+9.45}$ | $\mathbf{76.83}_{+8.19}$ | $\mathbf{38.4}_{+4.6}$ | $\mathbf{43.22}_{+5.41}$ | $\mathbf{23.11}_{+3.07}$ | $\mathbf{26.62}_{+5.94}$ | $7.34_{-0.24}$ | $8.34_{-0.19}$ |
| SCL-EXP | $\mathbf{75.9}_{+11.04}$ | $\mathbf{75.75}_{+10.35}$ | $\mathbf{40.95}_{+6.36}$ | $\mathbf{41.63}_{+4.67}$ | $\mathbf{24.96}_{+5.56}$ | $\mathbf{26.64}_{+5.61}$ | $7.21_{-0.34}$ | $\mathbf{8.47}_{+0.36}$ |
| L-W | $\mathbf{67.2}_{+10.99}$ | $\mathbf{71.07}_{+11.89}$ | $\mathbf{33.89}_{+5.85}$ | $\mathbf{38.16}_{+3.61}$ | $\mathbf{22.28}_{+7.93}$ | $\mathbf{23.19}_{+4.08}$ | $\mathbf{7.58}_{+0.5}$ | $8.64_{-0.1}$ |
| L-UW | $\mathbf{72.39}_{+11.51}$ | $\mathbf{73.26}_{+10.83}$ | $\mathbf{34.61}_{+3.98}$ | $\mathbf{40.3}_{+5.17}$ | $\mathbf{23.31}_{+7.3}$ | $\mathbf{24.41}_{+4.99}$ | $\mathbf{7.47}_{+0.11}$ | $\mathbf{8.96}_{+0.25}$ |
| PC-sigmoid | $\mathbf{45.72}_{+17.52}$ | $\mathbf{46.53}_{+7.24}$ | $\mathbf{33.24}_{+8.86}$ | $\mathbf{40.72}_{+4.84}$ | $\mathbf{12.81}_{+3.09}$ | $13.84_{-2.61}$ | $\mathbf{14.15}_{+4.88}$ | $\mathbf{17.06}_{+2.8}$ |
| MAE | $\mathbf{61.26}_{+3.89}$ | $\mathbf{63.41}_{+4.91}$ | $\mathbf{21.74}_{+5.44}$ | $\mathbf{23.65}_{+4.21}$ | $\mathbf{20.03}_{+3.41}$ | $\mathbf{21.79}_{+4.16}$ | $\mathbf{5.18}_{+0.07}$ | $\mathbf{6.68}_{+0.81}$ |

# I  Broader impacts

The datasets may advance the alorithms for learning from complementary labels. Those algorithms could learn a classifier with weak information. The privacy of the users may be easier to compromised as a result. We suggest the practitioners pay attention to the privacy issues when trying to utilize the collected datasets and the CLL algorithms.

# J  Access to the dataset and codes for reproduce

Please refer to the following link: `https://github.com/ntucllab/CLImage_Dataset`