# OpenReview forum: "CLImage: Human-Annotated Datasets for Complementary-Label Learning"
_NeurIPS.cc/2024/Datasets_and_Benchmarks_Track — Submitted to NeurIPS 2024 Track Datasets and Benchmarks_

### Official Review · Reviewer_dR23 · 2024-07-24

**Rating:** 5
**Confidence:** 3

**Review:**

The article aims to introduce the first Complementary-Label Learning (CLL) dataset. However, the logic of the article is convoluted, making it difficult to understand, and there are numerous writing errors present. Most importantly, the rationale for collecting a CLL dataset is not well-supported. The lack of a strong motivation and practical value casts doubt on the significance of this paper. The authors do not explain how is the real-world CLL dataset is a dataset of high-quality labels which is expensive and time-consuming to collect.
		To the best of my knowledge, the CL is employed to model training, rather than testing, so mostly labels are generated during the training process. Therefore, the question is, why do we need to have a real-world ground truth CL dataset?

**Strengths:**

1.	The article proposes a novel direction of collecting real-world datasets for Complementary-Label Learning (CLL) and discusses the potential value of these CLL datasets.

**Additional Feedback:**

no

**Clarity:**

a little difficult to read, both in terms of section organization and the support of examples within the sections.

**Correctness:**

The main claims and purposes claimed in the third section, datasets and goals, seem peculiar.

**Documentation:**

The main dataset and basic tutorial are released for reproduction.

**Ethics:**

Not suitable except research involving human subjects and its bringing bias.

**Limitations:**

1.	The paper lacks a strong motivation for collecting a CLL dataset, necessitating a clearer explanation of its necessity and relevance to the field.
2.	The logical structure is difficult to comprehend, for instance, why does the "results analysis" appear before the "experiments" section? Could it be part of the experiments?
3.	The presentation is not clear.

**Opportunities For Improvement:**

1.	Test the real-world CLL dataset, and show that the dataset is great to be used training effective classifiers. (See the 1st passage in Section Introduction)
2.	A strong motivation for collecting a real-world CLL dataset.
3.	The section Results Analysis seems to be important, and a clear presentation could be more significant for the motivation of real-world CLL dataset and the observations. What are the two related figures trying to say?

**Relation To Prior Work:**

The authors claim that this is the first CLL dataset. They do not compare it to other dataset of multi-class classification, other than try to be supported by the following cited papers in Line 53.
	[1] W.-Y. Lin. Reduction from complementary-label learning to probability estimates. Master’s thesis, 2023.
	[2] W. Wang, T. Ishida, Y.-J. Zhang, G. Niu, and M. Sugiyama. Learning with complementary labels revisited: The selected-completely-at-random setting is more practical.
	[3] W. Wang, T. Ishida, Y.-J. Zhang, G. Niu, and M. Sugiyama. Learning with complementary labels revisited: A consistent approach via negative-unlabeled learning. arXiv preprint arXiv:2311.15502, 2023.
	[4] Y. You, J. Huang, B. Wang, and Q. Tong. Rethinking one-vs-the-rest loss for instance-dependent complementary label learning.

**Summary And Contributions:**

The article proposes that experiments have revealed a significant performance decline when transitioning from synthetic datasets to real-world datasets. Further experiments indicate that annotation noise is the most influential factor affecting the annotated datasets.

---

> ### Author Rebuttal · Authors · 2024-08-17
>
> Thank you for your valuable questions and insightful comments. Please find our responses below.
>
> **1**. The paper lacks a strong motivation for collecting a CLL dataset, necessitating a clearer explanation of its necessity and relevance to the field.
>
> **Answer**:
>
> Thank you for giving us a chance to illustrate our motivation more clearly. Before our work, previous studies [1], [2], [3], [4], [5], [6], [7], [8] primarily relied on synthetic datasets, often created under assumptions of uniformity, bias, or noise. Our key motivation is to answer whether the assumptions hold true in the real-world data collection protocol. Our work clearly answers the question, shedding lights for the future development of CLL solutions. In particular, the results that we obtained are very different from those obtained with synthetic data. While we believe that CLL has its potential in trading expensive expert labels with cheaper human labels in some applications, we admit that the field has not moved to such applications yet, as the modeling side arguably needs to make CLL solutions better performing through fundamental studies before the application side wants to try them. That is our main purpose of curating the human-annotated datasets—assisting the fundamental studies. We believe that we are making an important and necessary step for the many theoretically-derived models to transit to future CLL applications. The four papers that came rapidly after our datasets, with three of them being outside our team, strongly support the relevance and impact of our work to the field. We will improve our wording in the revision to make the motivations above more clear to the readers.
>
> **2**: The logical structure is difficult to comprehend, for instance, why does the "results analysis" appear before the "experiments" section? Could it be part of the experiments?
>
> **Answer**:
>
> Thank you for pointing this out. The original "Result Analysis" in Section 4 was designed to closely examine the characteristics of our CLImage dataset before running the experiments on testing the accuracy of the solutions. We believe that it is necessary to keep it separate and before the “Experiments” section, but we agree that there is a potential confusion because of the section title. In response, we will change the title of Section 4 “**Dataset Characteristics**” in the revision to reflect our purpose. While the majority of the reviewers kindly appraised the paper’s clarity, we will strive to keep improving the clarity with more editorial actions.
>
> **3**: The authors claim that this is the first CLL dataset. They do not compare it to other dataset of multi-class classification
>
> **Answer**:
>
> Thank you for recognizing that this is the first CLL dataset—or to state more clearly, the first human-annotated CLL datasets. They connect with the rich literature of their ordinary multi-class classification ancestors and we have done comparisons to them with the row of “standard supervision” in the tables. It is not fully clear to us what you mean by “comparing with other dataset of multi-class classification.” But similar to how we answered the other reviewer, we agree that extending the dataset collection protocol and benchmarking to other datasets is a very important task. We chose, as the first step, to start with natural image datasets because (a) Those datasets, in their synthetic form, have been intensively used by existing works [1], [2], [3], [4], etc.. In fact, more than 90% of CLL papers are based on natural image datasets *without human-annotated complementary labels*, and our careful creation of human-annotated versions allow immediate fair comparison with those; (b) compared with more sophisticated datasets like medical imaging, those are easier to label by crowd-sourcing instead of requiring expert annotators. We hope that our successful first attempt can encourage the field (including ourselves) to extend the collection protocol to other domains. We will include the discussion in the revision.
>
> **References**:
>
> [1] Lin, WI., Lin, HT. (2023). Reduction from Complementary-Label Learning to Probability Estimates. In Advances in Knowledge Discovery and Data Mining. PAKDD 2023. vol 13936.
>
> [2] Yasuhiro Katsura and Masato Uchida. Bridging ordinary-label learning and complementary-label learning. In Proceedings of The 12th ACML, pages 161–176, 2020.
>
> [3] Takashi Ishida, Gang Niu, Aditya Menon, and Masashi Sugiyama. Complementary-label learning for arbitrary losses and models. In Proceedings of the 36th ICML, pages 2971–2980, 2019.
>
> [4] Yu-Ting Chou, Gang Niu, Hsuan-Tien Lin, and Masashi Sugiyama. Unbiased risk estimators can mislead: A case study of learning with complementary labels. In Proceedings of the 37th ICML, pages 1929–1938, 2020.
>
> [5] Y. Gao and M.-L. Zhang. Discriminative complementary-label learning with weighted loss. In International Conference on Machine Learning, pages 3587–3597. PMLR, 2021.
>
> [6] T. Ishida, G. Niu, W. Hu, and M. Sugiyama. Learning from complementary labels. Advances in neural information processing systems, 30, 2017.
>
> [7] H. Ishiguro, T. Ishida, and M. Sugiyama. Learning from noisy complementary labels with robust loss functions. IEICE TRANSACTIONS on Information and Systems, 105(2):364–376,2022.
>
> [8] D.-B. Wang, L. Feng, and M.-L. Zhang. Learning from complementary labels via partial-output consistency regularization. In IJCAI, pages 3075–3081, 2021.

---

> > ### Comment · Reviewer_dR23 · 2024-09-01
> >
> > the rebuttal solves my concerns. I increase my score to 6

---

> > > ### Author Response · Authors · 2024-09-01
> > >
> > > Thank you so much for your response, **Reviewer dR23**.
> > > We really appreciate your valuable comments, which help us improve our paper better and thanks for your re-evaluation!

---

> > > > ### Author Response · Authors · 2024-09-01
> > > >
> > > > Dear Reviewer dR23,
> > > >
> > > > We have a question regarding the new score; it seems the new score has not been updated yet.
> > > > We wonder if we need to make any requests or ask for any support from the workflow chair to update it.
> > > >
> > > > Thank you so much.

---

### Official Review · Reviewer_e9TQ · 2024-07-25
**A paper with useful datasets for CLL**

**Rating:** 6
**Confidence:** 3
**Correctness:** Yes
**Clarity:** Yes

**Review:**

This paper points out a problem of all prior work in CLL, that the datasets used for evaluation have been primarily synthetic datasets, and it's unclear how noisy the real-world datasets are and how do they affect the performance of current CLL algorithms.

The authors then collected four complementary-labeled datasets to bridge this gap, and conducted multiple experiments which revealed significant performance gap between their human annotated dataset and previous synthetically generated dataset.

Although the authors claim these to be "real-world" dataset, I wonder if there can be any bias introduced by the human annotators on Amazon Mechanical Turk due to the process setup, and if this process can really mimic what "real-world" datasets are in this field. It could be better explained with an example, e.g., what a real-world scenario that requires CLL would look like, and how the dataset collected in the paper can reflect the real-world setting.

Moreover, one interesting note mentioned in the paper is that the authors stopped to collect dataset for CIFAR100 because preliminary testing revealed all SOTA CLL algorithms performed poorly on that. However, this can still be useful to promote future CLL research, as a challenge to conqueror, which the authors can consider.

Lastly, what are the future research directions in CLL identified through these new benchmark experiments?

**Strengths:**

This paper contributes four human-annotated datasets for CLL research, and conducted multiple experiments on them which revealed additional insights for CLL.

**Additional Feedback:**

None

**Documentation:**

Yes

**Limitations:**

Yes

**Opportunities For Improvement:**

Clearly states how the data collection protocol reveal real-world data setting, starting from what real-world scenario would require CLL.

Consider to collect larger-scale dataset too to promote future research.

**Relation To Prior Work:**

Yes

**Summary And Contributions:**

This paper identifies the data insufficiency issue in CLL - where prior papers only use synthetic datasets, and generates multiple "real-world" datasets for CLL.

---

> ### Author Rebuttal · Authors · 2024-08-17
>
> Thank you for your valuable questions and insightful comments. Please find our responses below.
>
> **1**. Clearly states how the data collection protocol reveal real-world data setting, starting from what real-world scenario would require CLL.
>
> **Answer**:
>
> Thank you for the suggestion. We believe that our simple data collection protocol is likely feasible for any real-world multi-class classification application. The collection protocol requires the K classes to be known, which is typical for ordinary multi-class classification. It then randomly picks 4 out of K classes for the human annotator to select one “INCORRECT” label. We choose to start with the simple protocol instead of designing more sophisticated ones to keep the general and broad applicability. While we believe that CLL has its potential in trading expensive expert labels with cheaper human labels in some applications, we admit that the field has not moved to such applications yet, as the modeling side arguably needs to make CLL solutions better performing through fundamental studies before the application side wants to try them. That is our main purpose of curating the datasets—assisting the fundamental studies. We will clarify the above in the revision.
>
> **2**. Consider to collect larger-scale dataset too to promote future research.
>
> **Answer**:
>
> Thank you for the suggestion. Even before we started collecting CLCIFAR20, we did attempt to collect CLCIFAR100, which has 100 classes. We then realize that the CLL performance on the collected data is dreadful, being very close to random guesses. This dreadful level of performance persists even when using synthetic (uniform and noiseless) complementary labels on CIFAR100. The accuracies of FWD and SCL-NL, two state-of-the-art methods, are only slightly better than random guess (1% accuracy).
>
> | Method   | ResNet18 | ResNet34 | ResNet50 | DenseNet |
> |----------|----------|----------|----------|----------|
> | SCL-NL   | 1.42%    | 1.63%    | 1.19%    | 1.36%    |
> | FWD-R    | 3.44%    | 1.51%    | 3.10%    | 4.21%    |
>
> We thus decided to start building our collection with 10 and 20 classes, which are the typical scale used by current CLL works [1] [2] [3] [4], to pave the field’s way towards more classes. We totally agree that we should continue the efforts on both the dataset side and the modeling side to push for more classes. We will illustrate more about the challenges that we faced in the revision to help the field move to more classes.
>
> **References**:
>
> [1] Lin, WI., Lin, HT. (2023). Reduction from Complementary-Label Learning to Probability Estimates. In Advances in Knowledge Discovery and Data Mining. PAKDD 2023. vol 13936.
>
> [2] Yasuhiro Katsura and Masato Uchida. Bridging ordinary-label learning and complementary-label learning. In Proceedings of The 12th ACML, pages 161–176, 2020.
>
> [3] Takashi Ishida, Gang Niu, Aditya Menon, and Masashi Sugiyama. Complementary-label learning for arbitrary losses and models. In Proceedings of the 36th ICML, pages 2971–2980, 2019.
>
> [4] Yu-Ting Chou, Gang Niu, Hsuan-Tien Lin, and Masashi Sugiyama. Unbiased risk estimators can mislead: A case study of learning with complementary labels. In Proceedings of the 37th ICML, pages 1929–1938, 2020.

---

> > ### Comment · Reviewer_e9TQ · 2024-09-01
> >
> > Thanks for the follow-up! I don't have more questions.

---

> > > ### Author Response · Authors · 2024-09-01
> > >
> > > Thank you so much for your response, **Reviewer e9TQ**.
> > > We are glad to hear that our answer make these points clear for you.
> > > Once again, thank you so much for your insight comments!

---

### Official Review · Reviewer_6bMs · 2024-07-25
**Reviews from Reviewer 6bMs**

**Rating:** 6
**Confidence:** 3
**Correctness:** Yes.
**Clarity:** Yes.

**Review:**

### Strengths:

1. The paper introduces the first real-world datasets for CLL, which is a valuable step forward in validating the practical applicability of CLL algorithms.
2. The authors have designed a comprehensive protocol for collecting complementary labels from human annotators. This protocol is well-explained and appears to be robust, ensuring the collection of high-quality data.
3. The paper includes thorough benchmarking experiments and a detailed ablation study. The analysis of noise and bias in the collected datasets provides valuable insights into the challenges of real-world CLL.

### Weaknesses:
1. The paper assumes that human annotators provide complementary labels with less noise than random annotators. However, there is limited discussion on the variability and potential biases among different human annotators, which could impact the quality of the collected data.
2. The performance gap between real-world and synthetic datasets is attributed mainly to noise. However, a more nuanced analysis of other factors, such as dataset complexity or inherent dataset biases, could provide a more comprehensive understanding of this gap.

**Strengths:**

See detailed reviews.

**Additional Feedback:**

See weaknesses.

**Documentation:**

Yes.

**Ethics:**

No ethic issues.

**Limitations:**

Yes.

**Opportunities For Improvement:**

See weaknesses.

**Relation To Prior Work:**

Yes.

**Summary And Contributions:**

This paper addresses the problem of complementary-label learning (CLL) in a weakly-supervised learning paradigm, where the goal is to train a multi-class classifier using only complementary labels—labels that indicate the classes to which an instance does not belong. The authors identify a gap in the practical applicability of existing CLL algorithms due to their reliance on synthetic datasets and assumptions about the generation of complementary labels. To bridge this gap, the paper presents a protocol for collecting complementary labels from human annotators, resulting in four real-world datasets: CLCIFAR10, CLCIFAR20, CLMicroImageNet10, and CLMicroImageNet20. The authors conduct extensive benchmark experiments and a dataset-level ablation study, revealing a significant performance drop when transitioning from synthetic to real-world datasets, primarily due to annotation noise. The study underscores the need for CLL algorithms and validation schemes robust to noisy and biased complementary-label distributions.

---

> ### Author Rebuttal · Authors · 2024-08-17
>
> Thank you for your valuable questions and insightful comments. Please find our responses below.
>
> **1**. The paper assumes that human annotators provide complementary labels with less noise than random annotators. However, there is limited discussion on the variability and potential biases among different human annotators, which could impact the quality of the collected data.
>
> **Answer**:
>
> Thank you for pointing this out. To clarify, we observed, but did not assume, that human-annotated complementary labels are on average less noisy than random ones (noise rate 1/K). We analyze this in more detail below, showing the histogram of human annotators’ noise rate on the CLMIN10 dataset (*please refer the pdf file*). Results demonstrate that the majority of annotators are of nearly zero noise, while some annotators are similar to random-ones (noise rate 0.1), and some outliers are even worse than random ones (possibly because of misreading the guidelines or being adversarial). While we chose not to filter the higher-noise data to provide the community with the most realistic collection, the findings are very interesting and may provide guidelines for collecting better complementary labels in the future. We will add the findings, including deeper analysis on individual human bias, to the Appendix of the revision.
>
> **2**. The performance gap between real-world and synthetic datasets is attributed mainly to noise. However, a more nuanced analysis of other factors, such as dataset complexity or inherent dataset biases, could provide a more comprehensive understanding of this gap.
>
> **Answer**:
>
> We were able to attribute the performance gap between real-world and synthetic datasets to noise after a careful dataset-level ablation study (Section 5.3 and Appendix E) with data cleaning. This study reveals that by removing noisy data from the practical dataset, the performance gaps across all CLL algorithms are eliminated. Because the phenomenon is observed across all our current datasets, we did not have evidence on whether dataset complexity or inherent dataset biases also affect performance. We agree that those are potential future issues as we keep growing our collection to more datasets and will discuss this in the revision.

---

### Official Review · Reviewer_qMHj · 2024-08-06

**Rating:** 6
**Confidence:** 4
**Correctness:** Yes
**Clarity:** Yes

**Review:**

## Pros
- The paper addresses a significant gap in CLL research by providing real-world datasets, which helps in understanding the practical performance of CLL algorithms.

- The authors conducted thorough benchmark experiments, offering valuable insights into the performance of various CLL algorithms under real-world conditions.

-  The findings highlight the importance of developing CLL algorithms that are robust to noisy and biased label distributions, pointing the way for future research directions.

## Cons

- The current datasets have a limited number of classes. Authors argue that "state-of-the-art CLL algorithms cannot produce meaningful classifiers for 100 classes even on synthetic complementary labels that are uniformly and noiselessly generated". However, it is still necessary to conduct experiments to support these claims.

- While the datasets are derived from well-known classification datasets, it is unclear how generalizable the findings are to other types of data or domains.

**Strengths:**

- The authors developed a protocol for collecting complementary labels from human annotators, resulting in the creation of four unique real-world datasets that were not previously available.

- The authors conducted a meticulous analysis of the collected datasets, examining noise rates, label biases, and annotator behavior, which contributes to a nuanced understanding of CLL.

- The release of the datasets to the research community supports ongoing research and collaboration, fostering a collective effort to address the challenges identified in the paper.

**Additional Feedback:**

See **Opportunities For Improvement**

**Documentation:**

Yes

**Limitations:**

Yes

**Opportunities For Improvement:**

- Develop and test CLL algorithms on larger datasets with a greater number of classes to validate the claim regarding the limitations of current algorithms with 100 classes. This could involve creating new datasets or adapting existing larger-scale datasets for CLL.

- Conduct experiments to assess the generalizability of the findings across different domains and types of data. This could involve applying and testing the CLL algorithms and datasets in various fields such as medical imaging, natural language processing, or different areas within computer vision.

- Use state-of-the-art models and architectures that may offer better performance with complex datasets, to further test the limits and capabilities of CLL in real-world scenarios.

**Relation To Prior Work:**

Yes

**Summary And Contributions:**

The paper introduces a study on Complementary-Label Learning (CLL) that addresses the challenge of training classifiers with only negative labels indicating classes an instance does not belong to. The authors created four real-world CLL datasets using human annotations and discovered through benchmarking that performance significantly drops in real-world scenarios due to annotation noise and biases. The research emphasizes the need for robust CLL algorithms and validation methods, providing valuable insights and resources for the community.

---

> ### Author Rebuttal · Authors · 2024-08-17
>
> Thank you for your valuable questions and insightful comments. Please find our responses below.
>
> **1**. The current datasets have a limited number of classes. Authors argue that "state-of-the-art CLL algorithms cannot produce meaningful classifiers for 100 classes even on synthetic complementary labels that are uniformly and noiselessly generated". However, it is still necessary to conduct experiments to support these claims / develop and test CLL algorithms on larger datasets with a greater number of classes to validate the claim regarding the limitations of current algorithms with 100 classes. This could involve creating new datasets or adapting existing larger-scale datasets for CLL.
>
> **Answer**:
>
> Thank you for your suggestion. To the best of our knowledge, existing CLL methods have not been systematically tested on a greater number of classes, possibly because of its challenging nature. Our internal experiment results using synthetic (uniform and noiseless) complementary labels on CIFAR100 are listed below to support our claim that current state-of-the-art CLL algorithms struggle to produce meaningful classifiers for 100 classes. The accuracies of FWD and SCL-NL, two state-of-the-art methods, are only slightly better than random guess (1% accuracy). We will add this result to the paper to support our claim, and point the community towards this important future research direction.
>
> | Method   | ResNet18 | ResNet34 | ResNet50 | DenseNet |
> |----------|----------|----------|----------|----------|
> | SCL-NL   | 1.42%    | 1.63%    | 1.19%    | 1.36%    |
> | FWD-R    | 3.44%    | 1.51%    | 3.10%    | 4.21%    |
>
> **2**: Conduct experiments to assess the generalizability of the findings across different domains and types of data. This could involve applying and testing the CLL algorithms and datasets in various fields such as medical imaging, natural language processing, or different areas within computer vision.
>
> **Answer**:
>
> We agree that extending the dataset collection protocol and benchmarking to other domains is a very important task. We chose, as the first step, to start with natural image datasets because (a) Those datasets, in their synthetic form, have been intensively used by existing works [1], [2], [3], [4], etc.. In fact, more than 90% of CLL papers are based on natural image datasets *without human-annotated complementary labels*, and our careful creation of human-annotated versions allow immediate fair comparison with those; (b) compared with more sophisticated datasets like medical imaging, those are easier to label by crowd-sourcing instead of requiring expert annotators. We hope that our successful first attempt can encourage the field (including ourselves) to extend the collection protocol to other domains. We will include the discussion in the revision.
>
> **3**: Use state-of-the-art models and architectures that may offer better performance with complex datasets, to further test the limits and capabilities of CLL in real-world scenarios.
>
> **Answer**:
>
> Thank you for this great suggestion. We chose to start benchmarking with simpler models because overfitting is knowingly a serious issue for CLL [4] and simpler models may be less risky to overfitting. We are continuing to test the dataset on more sophisticated model architectures, as shown below. We will add the results to the revision.
>
> | Dataset    | Model  | SCL-NL  | FWD-R  |
> |-----|-----|-----|---|
> | CLCIFAR10  | ResNet18    | 34.77   | 38.12   |
> |    | ResNet34    | 25.13   | 27.60   |
> |    | ResNet50    | 24.55   | 26.06   |
> |    | DenseNet121 | 31.42   | 27.09   |
> | CLCIFAR20  | ResNet18    | 8.02    | 20.27   |
> |    | ResNet34    | 5.62    | 12.21   |
> |    | ResNet50    | 7.45    | 13.78   |
> |    | DenseNet121 | 7.98    | 14.26   |
> | CLMIN10    | ResNet18    | 21.80   | 30.15   |
> |    | ResNet34    | 12.60   | 15.20   |
> |    | ResNet50    | 10.40   | 21.00   |
> |    | DenseNet121 | 11.40   | 15.80   |
> | CLMIN20    | ResNet18    | 6.17    | 10.60   |
> |    | ResNet34    | 5.00    | 5.80    |
> |    | ResNet50    | 3.60    | 6.80    |
> |    | DenseNet121 | 5.10    | 6.50    |
>
>
> **References**:
>
> [1] Lin, WI., Lin, HT. (2023). Reduction from Complementary-Label Learning to Probability Estimates. In Advances in Knowledge Discovery and Data Mining. PAKDD 2023. vol 13936.
>
> [2] Yasuhiro Katsura and Masato Uchida. Bridging ordinary-label learning and complementary-label learning. In Proceedings of The 12th ACML, pages 161–176, 2020.
>
> [3] Takashi Ishida, Gang Niu, Aditya Menon, and Masashi Sugiyama. Complementary-label learning for arbitrary losses and models. In Proceedings of the 36th ICML, pages 2971–2980, 2019.
>
> [4] Yu-Ting Chou, Gang Niu, Hsuan-Tien Lin, and Masashi Sugiyama. Unbiased risk estimators can mislead: A case study of learning with complementary labels. In Proceedings of the 37th ICML, pages 1929–1938, 2020.

---

### Decision · Program_Chairs · 2024-09-26

**Decision:**

Reject

**Comment:**

Complementary-label learning (CLL) is a weakly-supervised learning paradigm that aims to train a multi-class classifier using only complementary labels, which indicate classes to which an instance does not belong. This paper develops a protocol to collect complementary labels from human annotators, and results in the creation of four datasets: CLCIFAR10, CLCIFAR20, CLMicroImageNet10, and CLMicroImageNet20. The experimental results showed a notable decrease in performance when transitioning from synthetic datasets to real-world datasets, where annotation noise is the most influential factor. In addition, this paper shows that the biased-nature of human-annotated complementary labels and the difficulty to validate with only complementary labels are two outstanding barriers to practical CLL. These findings suggest that the community focus more research efforts on developing CLL algorithms and validation schemes that are robust to noisy and biased complementary-label distributions.

This paper addresses a significant gap in CLL research by providing real-world datasets with extensive experiments. The overall contribution of this paper is important in the area of CLL.

After the rebuttal, all the reviewers gave the score of 6 (note that Reviewer dR23 indicated the intention to increase the score from 5 to 6). Therefore, I recommend accepting this paper as a poster presentation.

Note from PC: This year, the track has been incredibly competitive, which meant that many good papers had to be rejected. After careful discussion with the SACs we have concluded that this paper unfortunately cannot be accepted this time. This is the final decision, which cannot be appealed. We encourage the authors to incorporate feedback from reviewers and additional results / discussion provided during the author response period in their next submission.